# Two telomere-to-telomere gapless genomes reveal insights into *Capsicum* evolution and capsaicinoid biosynthesis

Weikai Chen[1,6], Xiangfeng Wang [1,6], Jie Sun [1,6], Xinrui Wang[1,6], Zhangsheng Zhu [1,2,6], Dilay Hazal Ayhan [1], Shu Yi[1], Ming Yan[1], Lili Zhang[1,3], Tan Meng[1], Yu Mu[1], Jun Li[1], Dian Meng[1], Jianxin Bian[1], Ke Wang[1,4], Lu Wang[1], Shaoying Chen [1], Ruidong Chen[1], Jingyun Jin[1], Bosheng Li [1], Xingping Zhang [1], Xing Wang Deng [1,5], Hang He [1,5] ✉ & Li Guo [1] ✉

Chili pepper (*Capsicum*) is known for its unique fruit pungency due to the presence of capsaicinoids. The evolutionary history of capsaicinoid biosynthesis and the mechanism of their tissue specificity remain obscure due to the lack of high-quality *Capsicum* genomes. Here, we report two telomere-to-telomere (T2T) gap-free genomes of *C. annuum* and its wild nonpungent relative *C. rhomboideum* to investigate the evolution of fruit pungency in chili peppers. We precisely delineate *Capsicum* centromeres, which lack high-copy tandem repeats but are extensively invaded by CRM retrotransposons. Through phylogenomic analyses, we estimate the evolutionary timing of capsaicinoid biosynthesis. We reveal disrupted coding and regulatory regions of key biosynthesis genes in nonpungent species. We also find conserved placenta-specific accessible chromatin regions, which likely allow for tissue-specific biosynthetic gene coregulation and capsaicinoid accumulation. These T2T genomic resources will accelerate chili pepper genetic improvement and help to understand *Capsicum* genome evolution.

Chili pepper (*Capsicum annuum*), a member of the nightshade family (Solanaceae), is a vegetable and spice crop that is cultivated worldwide and bears fruits best known for its fruit pungency, which is a result of capsaicinoids. Capsaicinoids are alkaloids that are synthesized via the convergence of the phenylpropanoid pathway and the branched-chain fatty acid pathway, followed by condensation via capsaicin synthase (CS)[1]; however, the full biosynthetic pathway has yet to be elucidated. The birth and death of capsaicinoid biosynthesis in the nightshade family are poorly understood by plant evolutionary biologists. The biosynthesis of capsaicinoids occurs in the fruit placental tissue of chili pepper, a unique trait of *Capsicum* spp., and is not detected in other Solanaceae plants, such as tomato[2]. However, fruit pungency is not universally present across *Capsicum*, as non-pungency is found in several cultivars and wild relatives (e.g., *Capsicum rhomboideum*) of chili peppers[3]. The mechanism by which *Capsicum* spp. gained and lost fruit pungency and how capsaicinoids are exclusively produced in fruits remain poorly understood due to the lack of high-quality genome resources available for the *Capsicum* genus, especially the nonpungent members.

The draft genome assembly of *C. annuum* (cultivar CM334) was released in 2014 and was 3.06 Gb in size with a contig N50 of 30 kb[2]. Afterward, the quality of several assemblies improved with the

[1]Peking University Institute of Advanced Agricultural Sciences, Shandong Laboratory of Advanced Agricultural Sciences in Weifang, Weifang 261325, China. [2]College of Horticulture, South China Agricultural University, Guangzhou 510642, China. [3]College of Modern Agriculture and Environment, Weifang Institute of Technology, Weifang 262500, China. [4]College of Life Sciences, Shandong Agricultural University, Tai'an 271018, China. [5]Peking-Tsinghua Center for Life Sciences, Peking University, Beijing 100871, China. [6]These authors contributed equally: Weikai Chen, Xiangfeng Wang, Jie Sun, Xinrui Wang, Zhangsheng Zhu. ✉e-mail: hehang@pku.edu.cn; li.guo@pku-iaas.edu.cn

development of single-molecule DNA sequencing technologies; for example, the assemblies of cultivar '59' (3.07 Gb, contig N50: 41.27 Mb)[4], 'Takanotsume' (3.05 Gb, contig N50: 99.05 Mb)[5] and CC-090 (3.06 Gb, contig N50: 187.09 Mb)[6]. To date, 23 genome assemblies of cultivated peppers, including *C. annuum, C. baccatum, C. chinense,* and *C. pubescens*, are publicly available[7–11], whereas genome sequences for wild peppers are very scarce[7]. Despite continuous improvement, the published assemblies still contain numerous gaps and assembly errors, and complete centromeres and telomeres are missing. Pepper genomes are repeat-rich (~80%), making genome assembly particularly challenging. Assembly gaps and errors often lead to mis-annotation of genes and the false discovery of genetic variants; thus, pepper functional genomic research remains challenging. Therefore, generating telomere-to-telomere (T2T) gapless and accurately annotated genome sequences is of paramount importance for improving the precise genetic characterization of peppers and dissecting the full biosynthetic pathways of capsaicinoids and other valuable natural products.

Accurate assembly of complete genome sequences remains a daunting task for eukaryotes with large and complex genomes with extensive repeats, high heterozygosity, or polyploidy. A breakthrough in assembling the complete human genome sequence, a milestone in human genomics, was recently achieved by the human T2T consortium[12]. This breakthrough has revolutionized the analysis of human genomic variants and epigenetic and transcriptional signatures in centromeres[13–15]. Nearly two decades after the publication of the first plant genome sequence, T2T gap-free genome assemblies were recently reported for *Arabidopsis*[16–18], rice[19,20], potato[21] and soybean[22]. However, these are considered near-complete genomes with either minor gaps in difficult-to-assemble regions or the omission of a few telomeres or centromeres that often contained high copies of tandem repeats. Furthermore, these published plant T2T genomes are relatively small (134 Mb[16] in *Arabidopsis*, 385 Mb[20] in rice, 773 Mb[21] in potato, and 1.01 Gb[22] in soybean). Recently, a complete genome assembly was reported for maize (2.10 Gb)[23]. However, complete genomes of large complex plant genomes, which are notoriously more difficult to assemble, are rare.

In this study, we perform de novo assembly and annotation of two T2T gapless *Capsicum* genome sequences, including a pungent pepper *C. annuum* and its nonpungent wild relative *C. rhomboideum*; these sequences constitute a milestone in pepper genome research. In-depth analysis of the two T2T genomes reveal distinctive structural, epigenetic, and transcriptional features in their centromeres. Evolutionary insights into the capsaicinoid biosynthesis pathway and regulation are obtained via phylogenomic and epigenomic data analyses. Our study provides timely genomic resources and insights, which will promote pepper research and genetic improvement.

## Results

### T2T gapless *Capsicum* genome assemblies

To assemble T2T gap-free genomes of pungent *C. annuum* and non-pungent *C. rhomboideum*, we generated high-coverage PacBio HiFi reads, Oxford Nanopore Technology (ONT) ultralong reads, Illumina paired-end (NGS) reads and high-throughput chromatin conformation capture (Hi-C) sequencing reads for the *C. annuum* double haploid cultivar G1-36576 and *C. rhomboideum* wild accession PI 645680 (Supplementary Fig. 1 and Supplementary Table 1). Genome assembly was performed using an in-house pipeline that integrates various computational tools to maximize the strength of various types of data (Supplementary Fig. 2 and Supplementary Table 2). Briefly, HiFi and ONT reads were first separately assembled using hifiasm[24] and NextDenovo[25], respectively. The *C. annuum* HiFi-based assembly was 3.13 Gb with a contig N50 of 262.4 Mb, containing 18 telomeres with thousands of copies of telomeric repeat units (TRUs) at one or both ends of 12 contigs, six of which were T2T. Assembling ONT reads generated a 3.10 Gb draft assembly with a contig N50 of 177.8 Mb,

containing 22 telomeres with more than 10,000 copies of TRUs, four of which were nearly T2T. ONT assembly was then used to fill the gaps (Supplementary Table 3) and patch telomeres in the HiFi assembly, yielding a hybrid assembly including 12 gapless chromosome-level contigs with 22 telomeres plus contigs containing 45S rDNA arrays. The nucleolus organizer regions (NORs) were separately assembled using 45S rDNA-containing HiFi reads, and the contigs were assembled into a single sequence based on the specific *k*-mer (Supplementary Fig. 3). To ensure accuracy, the sequences of ONT origin were replaced with their corresponding HiFi assembled contigs, followed by Hi-C scaffolding to 12 chromosomes (Fig. 1a) and manual correction for misassemblies using Juicebox[26]. After addition of the rDNA arrays and telomere patching, the final T2T gapless assembly of the *C. annuum* genome (CaT2T) was 3.1 Gb with a contig N50 of 262.6 Mb (Table 1); all 503 gaps in Ca59 were closed (Fig. 1b), and this complete *C. annuum* genome assembly represented the largest complete genome sequence reported so far (Fig. 1c). Using the same approach, we assembled a 1.70 Gb T2T gap-free genome sequence (CrT2T) of *C. rhomboideum* containing 13 chromosomes with a contig N50 of 146.0 Mb (Fig. 1a and Table 1), representing a gap-free genome for a non-domesticated *Capsicum* (Fig. 1c). We identified the complete set (24/24) of telomeres in *C. annuum* (Supplementary Fig. 4) and majority of (17/26) telomeres in *C. rhomboideum* (Supplementary Fig. 5). Synteny analysis revealed that 45.07% of the *C. rhomboideum* genes were syntenic to *C. annuum* (Supplementary Table 4), but their whole-genome alignment showed low sequence identity, suggesting substantial divergence. By comparing the two genomes, we demonstrated that at least 10 fissions and 11 fusions of chromosomes were required to obtain the *C. annuum* karyotype from that of *C. rhomboideum* (Fig. 1d and Supplementary Fig. 6).

### Genome validation and annotation

We performed extensive validations of the two T2T genome assemblies. First, we examined their Hi-C chromatin interaction maps, which revealed no obvious misplacement of contigs within the CaT2T and CrT2T assemblies (Supplementary Fig. 7). Then, we mapped all HiFi, ONT, and NGS reads separately against the assemblies, yielding a mapping rate of over 99.96% for all three data types (Supplementary Table 5). The mapped HiFi or ONT reads showed uniform coverage across the genome, with a few exceptions in CrT2T due to the presence of high-copy-number satellite repeats (Supplementary Fig. 7). CaT2T and CrT2T had quality values (QVs) of 56.60 and 77.18, respectively, and BUSCO scores of 98.62% and 97.12%, respectively, demonstrating the high accuracy and completeness of both assemblies (Table 1). Furthermore, aligning a recently published genome assembly of *C. annuum* cultivar '59' (hereafter Ca59)[4] against CaT2T assembly showed strong collinearity (Supplementary Fig. 2). The high-quality assembly of CaT2T was well supported by the high coverage of HiFi and ONT reads mapped in these gap regions (Supplementary Fig. 8 and Supplementary Table 6). Interestingly, we observed sporadic high-coverage read mapping against CaT2T and CrT2T, which corresponded to intact mitochondrial or chloroplast genomes; this result was validated by the high coverage of ONT ultralong read mapping spanning the entire integration site (Supplementary Fig. 9 and Supplementary Table 7), suggesting recent plastid genome integration in the nuclear genome.

Repeat annotation revealed that 79.5% (2.45 Gb) and 74.6% (1.28 Gb) of the *C. annuum* and *C. rhomboideum* genomes were repetitive sequences, primarily composed of transposable elements (TEs), especially long terminal repeat (LTR) retrotransposons (Supplementary Table 8). While LTR insertion in *C. rhomboideum* occurred relatively recently, *C. annuum* had two bursts of insertion approximately 0.1 million years ago (Mya) and 3.9 Mya (Supplementary Fig. 10), consistent with previous report on Ca59 assembly[4]. Both pepper genomes had low contents of satellite repeats (<0.01%), much fewer

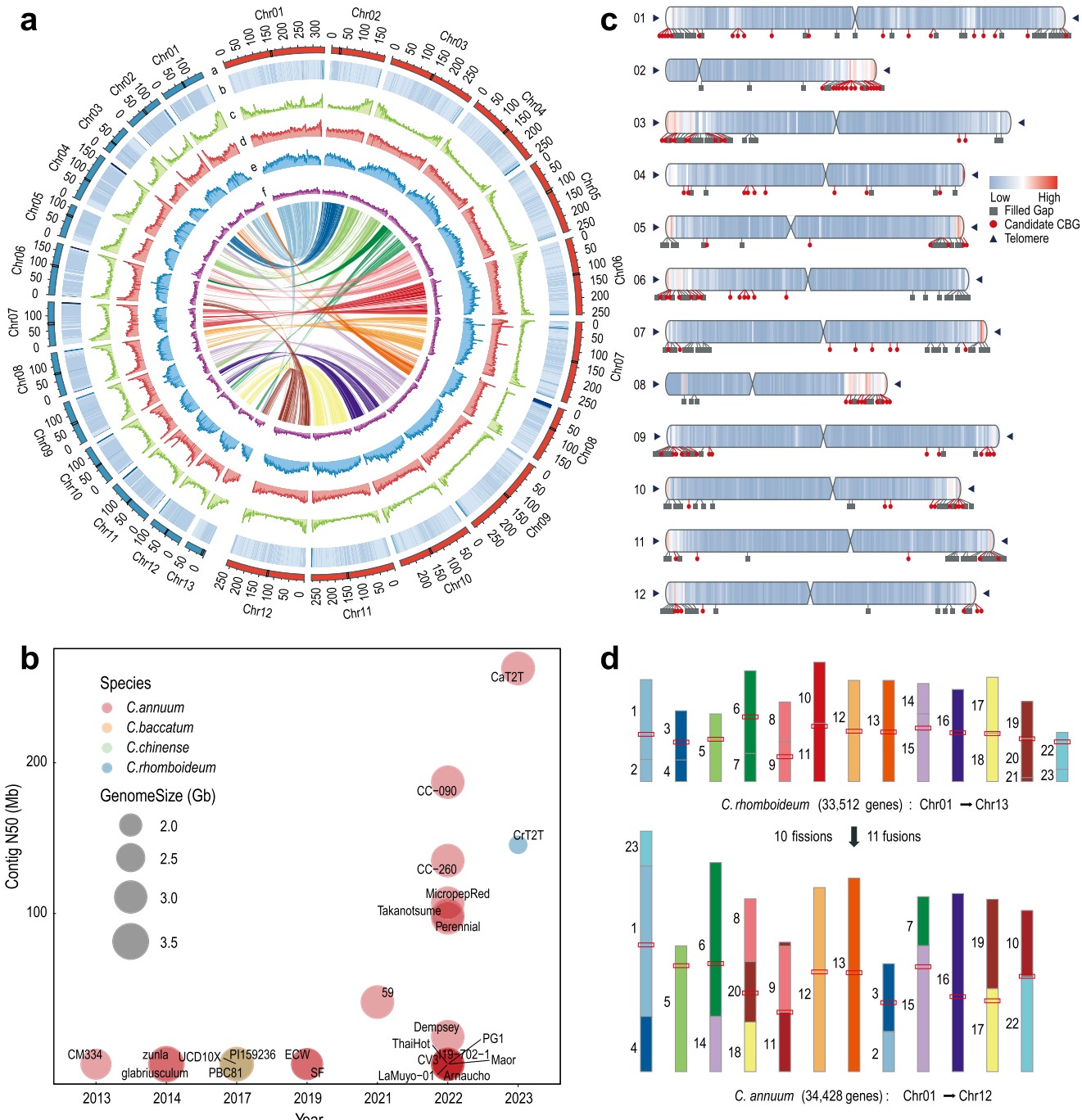

**Fig. 1 | T2T gap-free assembly of two *Capsicum* genomes. a** Circos plot showing the T2T genome assemblies for *C. annuum* (CaT2T) and *C. rhomboideum* (CrT2T). The track from a to g is as follows: chromosomes (Red: Ca, Blue: Cr), GC content, gene density, TE density, LTR/*Gypsy* density, LTR/*Copia* density and color ribbons representing genome-wide syntenic blocks. Centromeric regions (black) are denoted on the chromosome tracks. **b** A bubble plot highlighting the key statistics of the *Capsicum* genome assemblies in this study (CaT2T and CrT2T) and those reported previously. **c** A chromosome ideogram of the CaT2T genome assembly showing the mapping of filled gaps, candidate capsaicin biosynthesis genes (CBGs), centromeres and telomeres, with a heatmap of gene density overlaid on each chromosome. **d** Reconstruction of the rearrangement (fissions and fusions) events between CaT2T and CrT2T. Synteny blocks were colored by MCScanX. The red rectangles denote the centromere regions. Source data are provided as a Source Data file.

than those of humans (4.5%), the model plant *Arabidopsis* (0.37%), and its Solanaceae relative tobacco (1.75%). A total of 34,428 and 33,512 protein-coding genes were predicted for CaT2T and CrT2T, respectively, by using a combination of ab initio prediction, homologous proteins, and transcriptomic data. The gaps filled by CaT2T encoded 614 genes, 110 of which were newly annotated (nonsyntenic to Ca59) (Supplementary Fig. 11). Both *Capsicum* genomes were gene-dense toward the ends of chromosomes but gene-sparse toward

centromeres. CaT2T encodes 117 putative capsaicinoid biosynthesis genes (CBGs), including previously reported genes encoding capsaicin synthase (CS), aminomethyltransferase (AMT), ketoacyl-ACP synthase (Kas) and acyl carrier protein (ACL)[27] (Supplementary Data 1). These putative CBGs were expressed in at least one tissue, and 26 of them showed much higher expression levels in the fruit placenta than in other tissues (Supplementary Fig. 12); therefore, serving as strong candidates for full elucidation of capsaicinoid biosynthetic enzymes.

**Table 1 | Statistics for genome assembly and annotation of two pepper species**

| Genomic feature | *Capsicum annuum* | *Capsicum rhomboideum* |
|---|---|---|
| Number of contigs | 12 | 13 |
| Total length (bp) | 3,103,116,129 | 1,707,653,203 |
| Contig N50 (bp) | 262,573,928 | 145,987,823 |
| Number of gaps | 0 | 0 |
| Number of telomeres | 24 | 17 |
| Number of centromeres | 12 | 13 |
| Number of gene models | 34,428 | 33,512 |
| GC content (%) | 35.00 | 36.31 |
| Repeat content (%) | 79.50 | 74.64 |
| Assembly BUSCOs (%) | 98.62 | 97.12 |
| Annotation BUSCOs (%) | 97.04 | 93.23 |
| QV | 56.60 | 77.18 |
| Completeness (%) | 96.49 | 98.11 |

## Capsicum centromeres are extensively invaded by CRM retrotransposons

Centromeres, which are essential for faithful chromosomal segregation during cell division, are typically heterochromatic regions with megabase arrays of tandem repeats where the kinetochore protein complex binds[28]. We first identified the centromeres of CaT2T by generating CENH3 ChIP-seq data for *C. annuum*, which clearly delineated the locations and boundaries of the 12 centromeres in CaT2T (Fig. 2a). Then, we observed that the interchromosomal interactions were always positively correlated with the ChIP-seq peak, especially in CrT2T (Supplementary Fig. 13). Unlike the *Arabidopsis*[16] and human[13] centromeres, *Capsicum* centromeres lacked high-copy tandem satellite repeats and higher-order repeats (HORs), which is suggestive of newly formed centromeres[29]. Moreover, we found that *Capsicum* centromeres were extensively invaded by *Gypsy*-LTRs, accounting for ~71% of the total centromeric sequences (Fig. 2b), with *Gypsy*-LTRs accounting for only 47.3–49.7% of the whole genome (Supplementary Table 8). This pattern was also reported in einkorn wheat[30] and cotton[31], where more than 80% of their functional centromeres were *Gypsy*-LTRs. We also found that the burst of LTR insertion in centromeres was later than that in the whole genome, indicating that the recent evolution of centromeres was shaped by LTR insertion (Fig. 2b). LTRs are typically subjected to reshuffling and rearrangement due to unequal homologous recombination events that generate fragmented or solo LTRs[32]. A number of solo LTRs and intact LTRs were identified in the two *Capsicum* genomes (Supplementary Table 9). We observed that *Capsicum* centromeres had a weaker ability than non-centromeres to remove LTRs, as indicated by their significantly lower solo-to-intact LTR ratios than those of the whole genome (Fig. 2c). Synteny analysis revealed that centromeric retrotransposons of maize (CRMs) were enriched in centromeres of several *Capsicum* genomes (Supplementary Fig. 13), suggesting that the distribution of CRMs was a marker that could be used to identify *Capsicum* centromeres without relying on ChIP-seq data. CRMs possess chromodomain or CR motifs that interact with centromeric histones and play important roles in centromere evolution and function[33]. Furthermore, phylogenetic analysis of *Gypsy*-LTR retrotransposons in two *Capsicum* and potato genomes revealed six subfamilies with two major subfamilies: *Athila* and *Tekay*. *Athila* LTRs are the major LTRs in *Arabidopsis* centromeres[16], unlike *Capsicum* centromeres, which are enriched with CRM *Gypsy*-LTRs (Fig. 2d; Supplementary Table 9; Supplementary Data 2). The lack of satellite repeats and enrichment of CRM LTRs (Fig. 2e) distinguished *Capsicum* centromeres from centromeres reported in other published plant T2T genomes[16,20–22]. Strikingly, CrT2T had a greater content of CRMs than

CaT2T, which likely contributed to the distinctive signature of sequence identity within its centromeres (Supplementary Fig. 14). For both T2T genomes, we observed low interspecies and interchromosomal centromere sequence identity, suggesting rapid divergence of *Capsicum* centromeres within and between species; this result is consistent with the findings of a recent study of *Arabidopsis* centromeres[34].

## Centromeres and telomeres are transcriptionally and epigenetically active

Centromeres and telomeres are poorly understood genomic regions in terms of protein-coding genes and transcriptional and epigenetic control. Genome annotation revealed that 60 genes in CaT2T centromeres were enriched in functions such as response to freezing, changes in DNA topology and meiotic chromosome separation (Supplementary Fig. 15). In contrast, CrT2T centromeres encoded 94 genes enriched in the response to UV-B, photosynthesis and the regulation of circadian rhythm. Interestingly, only six or seven centromere genes were homologous (Supplementary Data 3). The low homology of centromere genes reflected the high divergence of centromeres between the species, consistent with their poor whole-genome sequence alignment. We found that CaT2T centromeres (Fig. 3a and Supplementary Figs. 16 and 17) and telomeres (Fig. 3b) exhibited active transcription of both transposons and protein-coding genes, as suggested by RNA-seq analysis. For example, approximately 42 (70.0%) centromere-encoded genes were expressed in at least one tissue (TPM > 1) in *C. annuum*, including *CaT2T07g00954*, which encodes a telomere maintenance protein that protects the ends of telomeres from attack, and *CaT2T01g02835*, which encodes a protein that controls flowering time. The CrT2T centromere-encoded genes (92.5%) were more active with three tandem copies of regulator of chromosome condensation (*RCC1*) with an average TPM > 200; these genes could play key roles in the regulation of chromatin condensation in mitosis.

*Capsicum* genomes contain rich epigenomic signatures, such as DNA methylation, histone modifications, topologically associated domains (TADs) and A/B compartments[4]. However, little is known about these epigenetic marks in centromeres and telomeres. Therefore, we mapped our own generated (Hi-C and whole-genome bisulfite sequencing) and public epigenomic (histone ChIP-seq) data to the CaT2T assembly. Hi-C data analysis revealed A/B compartments, TADs, and small chromatin loops within centromeres (Fig. 3a). *C. annuum* centromeres primarily belonged to the "B" compartment, which is typically associated with low transcription; however, the "A" compartment was also detected on a few centromeres, such as on Chr08, Chr09 and Chr10 (Supplementary Fig. 17). Consistently, high TE density, low gene density (Fig. 3c), and low histone H3K9me2 ChIP-seq peaks were detected for heterochromatic centromeres and pericentromeres with enriched CENH3 ChIP-seq signals (Fig. 3d). Interestingly, although DNA methylation levels in centromeric and non-centromeric regions were comparable overall (Fig. 3c), we found that centromere-located CRMs showed lower gene-body CHG methylation than gene flanking regions (Fig. 3e and Supplementary Fig. 18), suggesting that CRMs have high transcriptional activity. Despite the nature of heterochromatin, transcription in centromeres was not silent, as we identified some highly expressed genes and TEs, such as one on Chr07 encoding the peroxiredoxin Q protein, which is involved in cell redox homeostasis (Fig. 3a). Compared to those of centromeres, most telomeres had low Hi-C mapping signals (Fig. 3b), perhaps due to the high density of tandem telomeric repeats. Subtelomere regions were relatively gene-rich, mostly associated with "A" compartments, and marked with low H3K9me3 ChIP-seq signals, except in gene-sparse regions (Fig. 3b). These results provide unprecedented insights into the organization and functions of complex genomic regions in pepper.

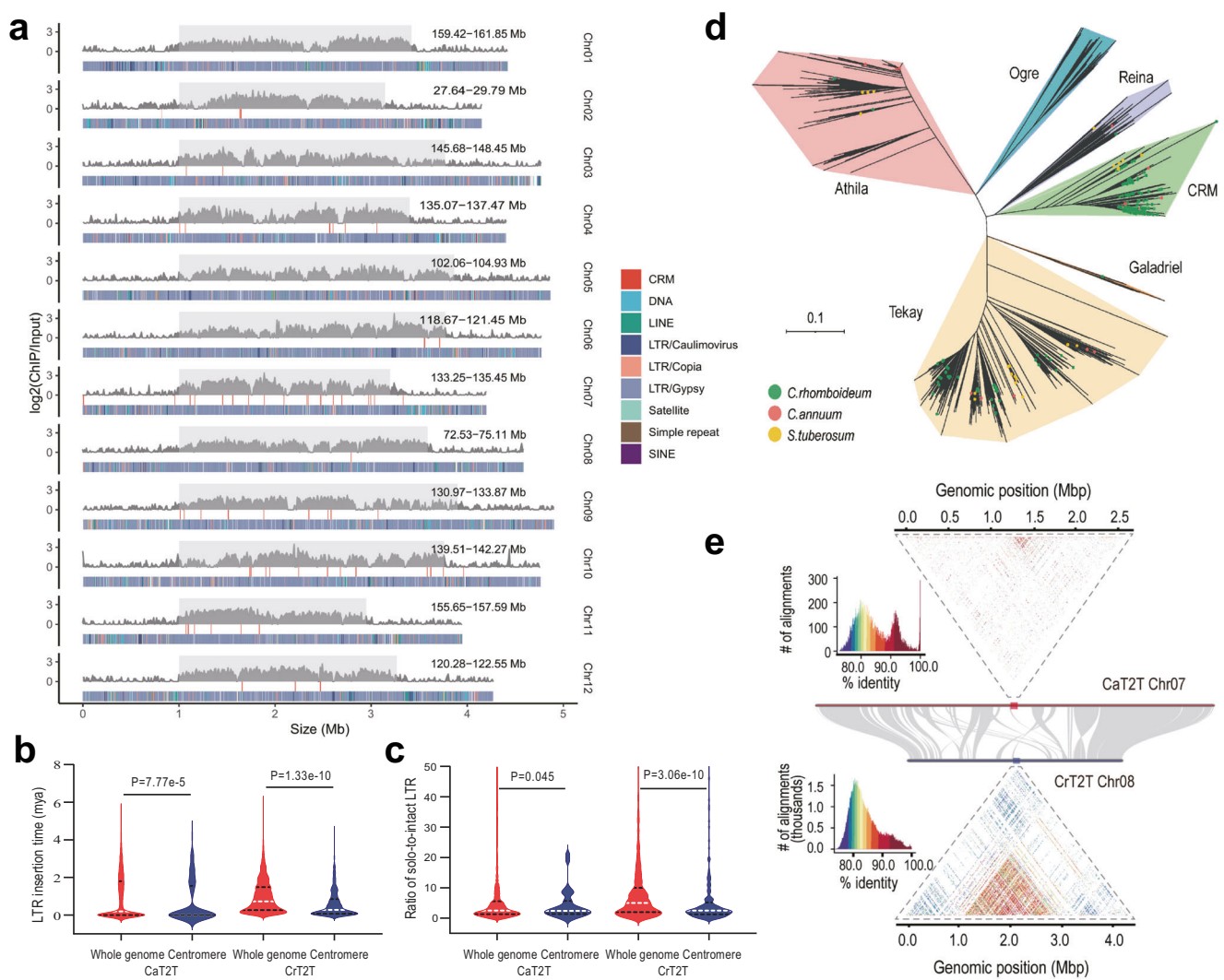

**Fig. 2 | Genome structure of centromeric regions in the *C. annuum* genome.**
**a** Schematic representation showing the distribution of different transposable elements across 12 centromeres in CaT2T. The CENH3 ChIP-seq signals (average of two biological replicates) are represented by the enrichment level in 30 kb windows and the ChIP-identified centromeres are marked by gray boxes. The red lines for the track of the CRM indicate that the intact CRM was located in the centromeres.
**b** Specific LTR insertion time distribution of the whole genomes and centromeres. The central white line and black line in the plot indicate the median and the upper and lower quartiles of insertion times respectively. Significant differences between

groups were assessed using the two-sided Wilcoxon rank-sum test. **c** Specific ratio of solo LTR to intact LTR-RT distributions in the whole genomes and centromeres. Significant differences between groups were assessed using the two-sided Wilcoxon rank-sum test. **d** Neighbor-joining trees constructed from genome-wide intact Ty3-Gypsy elements from *C. annuum*, *C. rhomboideum* and *S. tuberosum*. The red, green and yellow solid circles in the branches represent the elements located in centromeric regions. **e** An example of pairwise sequence identity of nonoverlapping 5 kb centromeric regions in *C. annuum* and *C. rhomboideum*. Source data are provided as a Source Data file.

## Evolutionary history of the capsaicinoid biosynthesis pathway in *Capsicum*

The mechanism underlying the initiation of capsaicinoid biosynthesis in plants remains poorly understood. Kim et al. first approached this question by comparing a fragmented pepper genome with a tomato genome, thereby revealing the mechanisms of pungency[2]. However, a better understanding of how the pathway emerged and evolved requires investigation in a broader phylogenetic context. Therefore, we revisited this question by performing phylogenomics using two T2T *Capsicum* genomes and 14 genomes of other angiosperms (Supplementary Table 10), including three pungent and 13 nonpungent species. We found that *Capsicum* was more related to *Physalis* (ground cherry) than to *Solanum* (*e.g.*, tomato), and diverged from the two taxa at -17 Mya and -19 Mya, respectively (Fig. 4a). The fact that capsaicinoid biosynthesis is limited to *Capsicum* indicated that specialized metabolites must have arisen in *Capsicum* after its divergence from *Physalis* at -17 Mya. Furthermore, *C. baccatum* diverged from

*C. annuum* and *C. chinense* at - 5 Mya, which together diverged from the nonpungent *C. rhomboideum* at -13.4 Mya, suggesting that the capsaicinoid pathway could have originated between 13.4 Mya and 5 Mya (Fig. 4a). To understand how the pathway arose in *Capsicum* plants, we identified genes from the 16 angiosperms with homologs to known CBGs using OrthoFinder[35] (Supplementary Data 4). All species, pungent or not, contained homologs of CBGs, except that *CS*, the most critical gene[1], appeared as tandem duplicates only in Solanaceae at approximately 71 Mya (Fig. 4b and Supplementary Fig. 19). In particular, *Capsicum* species had the most copies of *CS*, with seven in *C. annuum*, six in *C. chinense*, and four in *C. pubescens*, *C. baccatum* and *C. rhomboideum*. There were fewer copies in *C. pubescens* and *C. baccatum* than *C. annuum* due to either gene loss in the former two or tandem duplications in the latter. In fact, *CS* duplication was widely detected in Solanaceae (Fig. 4b), as previously reported in tomato[2], but tandem duplication mainly occurred in *Solanum*, *Physalis*, and *Capsicum* spp. between 30 Mya and 40 Mya (Fig. 4a). In addition,

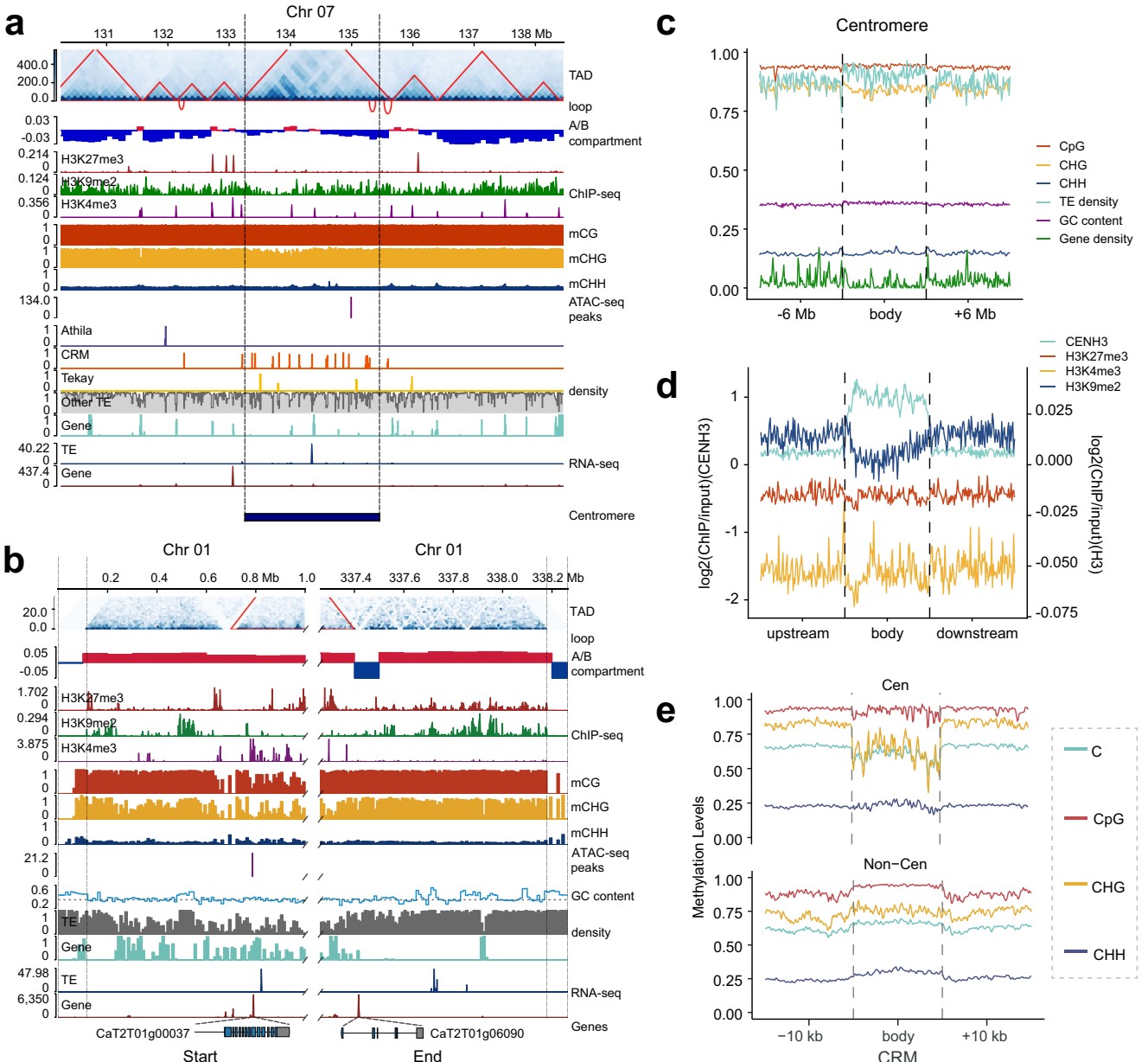

**Fig. 3 | Epigenomic and transcriptional features of difficult-to-access regions in the *C. annuum* T2T genome. a** Characteristics of the centromeres on *C. annuum* Chr07 as an example. The distributions of TADs, chromatin loops, A/B compartments, ChIP-seq signals (H3K27me3, red; H3K9me2, green; and H3K4me3, purple), methylation levels (CG, red; CHG, green; and CHH, purple), ATAC-seq signals, TE elements (*Athila*, blue; *CRM*, yellow; *Tekay*, red; and others, gray), genes (green), TE transcript abundances (blue) and gene transcript abundances (red) are plotted from top to bottom. **b** Epigenetic and transcriptional landscape of telomeres on *C. annuum* Chr01 as an example. **c** Epigenetic signals detected at or in proximity to centromeres with TE, GC and gene density. **d** CENH3 and histone modification ChIP-seq signals at or in proximity to centromeres. **e** The distribution of CG, CHG, and CHH methylation in the *Athila*, *CRM* and *Tekay* regions including the 10 kb upstream and downstream regions of *C. annuum*.

microsynteny analysis revealed that *CS* tandem duplicates were syntenic in *Capsicum, Physalis*, and *Solanum* (Fig. 4c), as were other CBGs (Supplementary Fig. 20). This result suggested that the expression of CBGs in nonpungent species may have been disrupted. Indeed, we found that many CBGs were highly expressed in the fruits of pungent *Capsicum* species (Fig. 4d), whereas the *C. rhomboideum* and *Physalis CS* and *KasI* genes were hardly expressed (Fig. 4d). Sequence alignment revealed that these syntenic *CS* copies (*CS-1/CS-2*) had conserved coding sequences (CDS) and upstream and downstream regulatory regions among pungent species, while nonpungent species had structural variations (SVs) within both the CDS and flanking regions (Fig. 4e and Supplementary Fig. 21). In addition, sequence variations

were observed in several other CBGs, including *ACL, BCAT, CCoAMT, FatA*, and *KasI* (Supplementary Fig. 20). The highly conserved *CS* copies were only present in cultivated *Capsicum*, suggesting that they were relatively recent (~5 Mya) tandem duplicates from older *CS* genes. Taken together, these results indicate that although nonpungent species contained homologs of functional CBGs, their coding and regulatory regions diverged significantly from those of *C. annuum*.

## Chromatin accessibility regulates tissue-specific capsaicinoid biosynthesis

Capsaicinoid biosynthesis is highly tissue specific and occurs only in fruits, particularly in the placenta, beginning at ~16 days post anthesis.

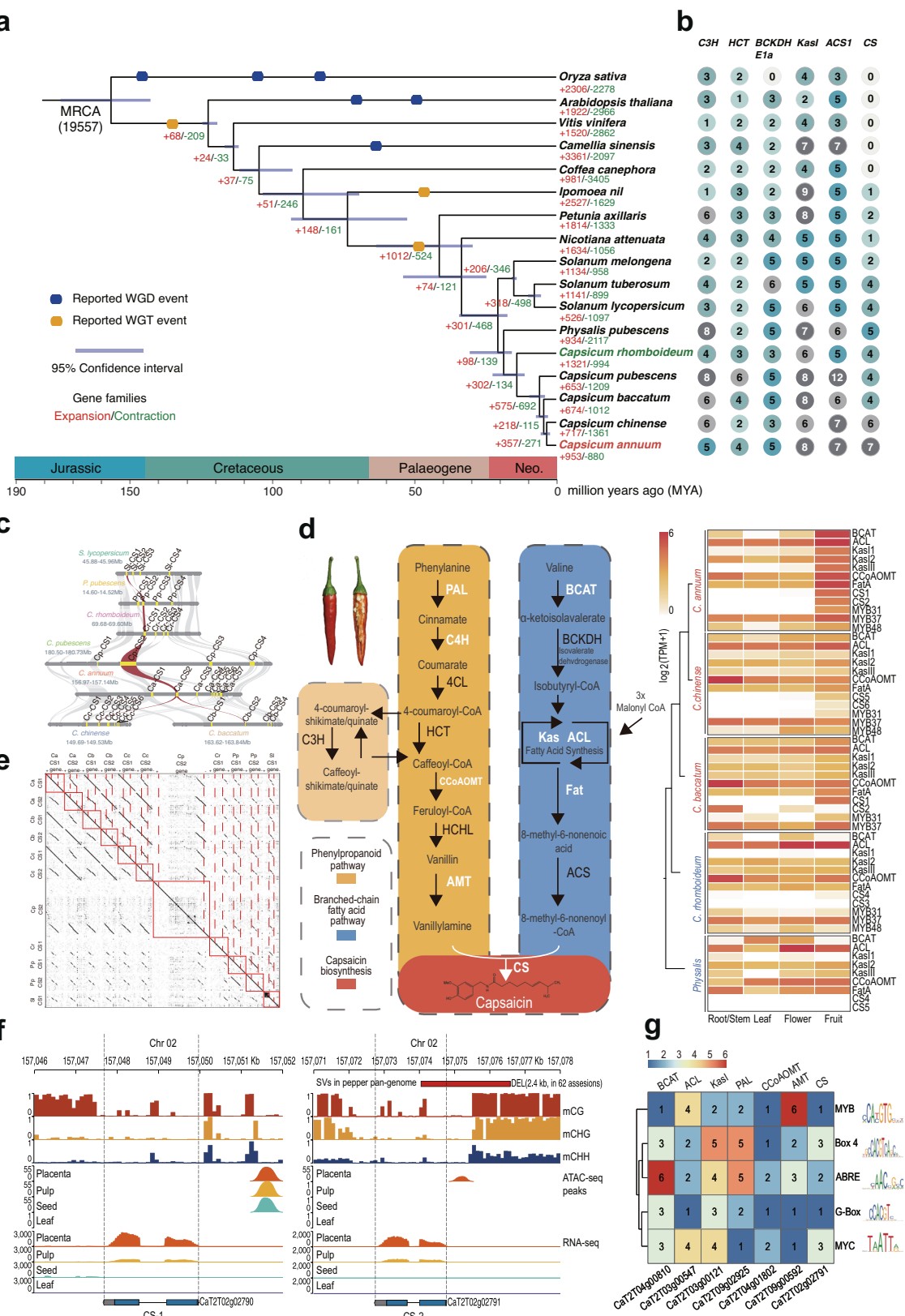

To understand how tissue specificity is achieved, we performed multiomic co-profiling of *C. annuum* fruits and leaves, including Assay for Transposase-Accessible Chromatin Sequencing (ATAC-seq), whole-genome bisulfite sequencing and RNA-seq, and analyzed the data using CaT2T as a reference. RNA-seq analysis revealed that *CS* and its transcriptional regulators *MYB31*[36] and *MYB48*[37] were specifically expressed in the placenta (Fig. 4f). Placenta-specific open chromatin regions (OCRs) with low methylation levels were detected within 2 kb upstream of *CS-2*, *MYB31* and *MYB48*, while *CS-1* also showed OCRs in both pulp and seeds, suggesting that *CS-2* is likely the primary functional gene that contributes to placenta-specific synthesis of capsaicinoids (Fig. 4f and Supplementary Fig. 22). Another 26 putative CBGs

**Fig. 4 | Evolution of capsaicin biosynthesis genes and their tissue-specificity.** **a**, **b** Phylogenomic analysis of *C. annuum* and related angiosperm species. Whole genome duplication (WGD) or triplication (WGT) events, and gene family expansion/contraction statistics are marked on the MCMC phylogenetic tree, which was constructed using single-copy orthologs (**a**). MRCA represents the most recent common ancestor. The panel alongside the species is a summary of the abundance of gene family members related to the biosynthesis of capsaicin in a phylogenomic context (**b**). **c** Microsynteny relationships of the capsaicin synthase (*CS*) gene and its tandem copies in the syntenic block, which are conserved among *Solanum lycopersicum* (Sl), *Physalis pubescens* (Pp), *C. annuum* (Ca), *C. rhomboideum* (Cr), *C. baccatum* (Cb), *C. chinense* (Cc) and *C. pubescens* (Cp). The red lines indicate the closest homologs of key *CS* gene. **d** Left: diagram of capsaicinoid biosynthesis pathways and key genes. PAL, phenylalanine ammonia-lyase; C4H, cinnamate 4-hydroxylase; 4CL, 4-coumarate: CoA ligase; HCT, hydroxycinnamoyl transferase;

CCoAOMT, caffeoyl-CoA 3-*O*-methyltransferase; C3H, coumarate 3-hydroxylase; HCHL, hydroxyl cinnamyl-CoA hydrase/lyase; AMT, aminotransferase; BCAT, branched chain amino acid aminotransferase; BCKDH, branched-chain α-ketoacid dehydrogenase; Kas, β-ketoacyl-ACP synthase; ACL, acyl carrier protein; FatA, acyl-ACP-thiesterase; ACS, acetyl-CoA synthetase; and CS, capsaicin synthase. Right: transcriptional expression heatmap of capsaicin biosynthesis genes and their homologous genes in different tissues of five Solanaceae plants. **e** Dot plot of the *CS* nucleotide sequence in seven Solanaceae plants. The plotted sequence includes full coding sequences of *CS* and tandem repeats as well as their 2 kb flanking sequences. **f** Epigenomic and transcriptomic patterns of two *CS* genes and flanking regions. **g** Common transcription factor binding motifs enriched in both the open chromatin region (ATAC-seq) and the upstream (2 kb) sequences of seven capsaicin biosynthesis genes. Source data are provided as a Source Data file.

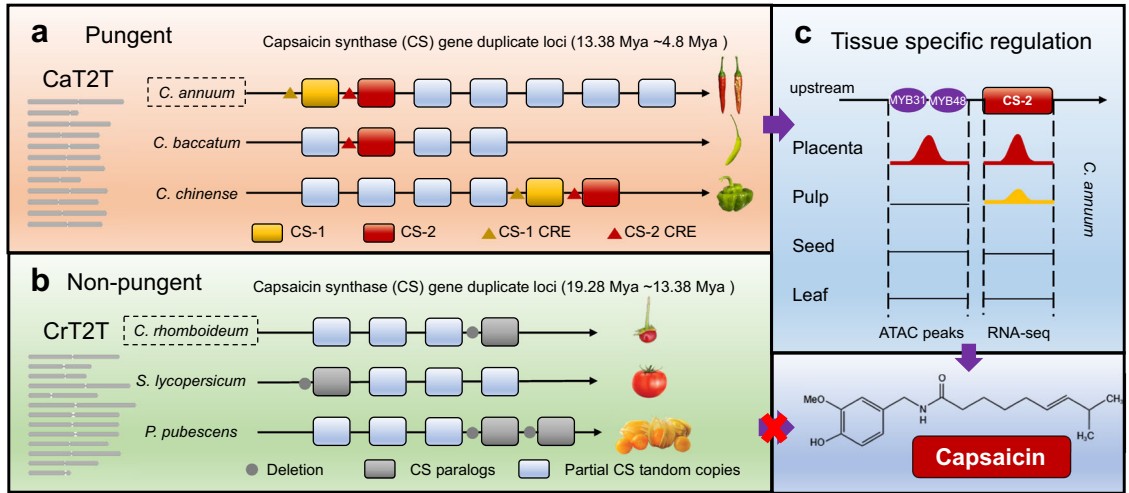

**Fig. 5 | Model for the evolution of capsaicin synthase (CS) in pungent and nonpungent Solanaceae plants. a** The pungent species *C. annuum* (CaT2T), *C. baccatum* and *C. chinense* all have at least one functional *CS* gene. In addition, *C. annuum, C. baccatum* and *C. chinense* have several tandem duplicates of *CS* genes (formed between 13.38 Mya and 4.8 Mya), most of which are partial gene paralogs and are non-functional. Mya: million years ago. CRE: *cis*-regulatory elements. **b** In contrast, the nonpungent tomato (*S. lycopersicum*), ground berry (*P. pubescens*), and *C. rhomboideum* (CrT2T) species lack a functional *CS* gene, but have various

numbers of *CS* tandem paralogs that include both partial *CS* or non-expressed *CS* due to deletion in CREs (formed between 19.28 Mya and 13.38 Mya). **c** Pungent peppers achieve tissue-specific regulation of capsaicin production through placenta-specific opening of the chromatin region around the *CS* gene (*CS2*), as identified by ATAC-seq, thus allowing for its transcription (determined via RNA-seq). In contrast, nonpungent species do not perform capsaicin biosynthesis due to the absence of either functional copies of *CS* genes or corresponding CREs.

encoding the phenylpropanoid and valine pathways were expressed in both fruits and leaves but lacked tissue-specific OCRs, suggesting that these genes function beyond capsaicinoid biosynthesis (Supplementary Fig. 12). Nonetheless, the increase in expression in the placenta compared to other tissues (Supplementary Fig. 12) indicated strong coregulation of capsaicinoid biosynthesis.

However, how coregulation of genomically dispersed CBGs is achieved to confer this tempo-spatial specificity of capsaicinoid production remains unknown. One would expect there should be common regulatory elements for CBGs. To validate this hypothesis, we extracted the placenta-associated OCRs identified by ATAC-seq and 2 kb upstream sequences of CBGs to perform sequence motif enrichment analysis, yielding a total of 38 enriched motifs ($p < 0.01$). Overlapping enriched motifs obtained from two enrichment analyses revealed five transcription factor binding motifs (TFBS), namely, MYB, G-box, Box-4, ABRE and MYC, which were present in all seven CBGs, including *CS, ACL, KasI, PAL, CCoAOMT* and *BCAT* (Fig. 4g). These significantly enriched TFBS within common OCRs were likely recognized by certain TFs, such as MYB31, which coregulated these CBGs in specific tissues[36]. In nonpungent *C. annuum* cultivars, the OCRs of *CS-2* were lost due to a 2.4 kb deletion, resulting in a lack of capsaicinoids in these accessions (Fig. 4f). In brief, multiomic data analysis illustrated a likely epigenetic mechanism for the tissue-

specific coregulation of capsaicinoid biosynthesis genes in chili pepper.

## Discussion

The two T2T gap-free *Capsicum* genome assemblies generated in this study represent key milestones in plant genome research. First, almost ten years after the publication of the first pepper genome[2], we mapped the complete genome sequence of *C. annuum*, which is the largest complete genome sequence reported thus far. The genomics of chili peppers has entered the era of complete T2T genomes, following the footsteps of humans and model plants such as *Arabidopsis*, rice and maize. Second, to investigate the mechanisms of pungency evolution, we assembled a T2T gapless genome for a nonpungent wild pepper *C. rhomboideum*. Through phylogenomics involving five *Capsicum* (four pungent, one nonpungent) and ten non-*Capsicum* (all nonpungent) genomes, we traced the evolutionary history of capsaicinoid biosynthesis pathways among closely related Solanaceae plants by estimating the times at which pungency emerged, the tandem duplications of key genes occurred, and the functional biosynthesis pathway developed in the evolutionary tree (Fig. 5). However, due to the lack of sufficient high-quality *Capsicum* genomes, in addition to the two assembled in this study, answering some key interesting evolutionary questions is difficult. For example, how did pungent *C.*

*baccatum* have one functional duplicate of *CS* genes, whereas non-pungent *C. rhombiodeum* managed to retain all four copies? Is it possible that functional *CS* genes and *cis*-regulatory elements were already present in Solanaceae plants but were later lost in nonpungent species? Alternatively, the five cultivated *Capsicum* species could gain functional *CS* genes or *cis*-regulatory elements, probably through natural mutations or TE transpositions during domestication. The *Capsicum* genus has more than 40 species with diverse genetic backgrounds and traits[38], but reference genomes are available for only four species thus far. With more genomes of *Capsicum* spp. with diverse capsaicin profiles available in the future, it will be possible to better infer when and how the emergence and loss of the pathway occurred.

Many plant secondary metabolites accumulate in multiple tissues and organs, but tissue-specific production of natural product molecules is not uncommon, with examples such as morphine in opium poppy capsules[39] and capsaicin in chili pepper fruits. The expression of biosynthetic genes in specific tissues is required for specificity, the regulatory mechanisms of which remain elusive. Typically, metabolic gene clusters can facilitate the coregulation of biosynthetic genes, as observed for morphine (poppy)[39] and thalianol (*Arabidopsis*)[40]. However, the genes involved in the biosynthesis of many plant metabolites, such as capsaicin and colchicine (lily)[41], are generally not clustered and dispersed. Overall, the spatial-temporal regulatory mechanisms of biosynthetic genes are poorly understood. Through comparative ATAC-seq and RNA-seq analysis of fruit and leaf tissues, we determined the fruit-specific open chromatin regions with several key biosynthetic genes and regulators encoding conserved TF binding sites (Fig. 5). This concerted gene regulation likely enabled efficient production of biosynthetic enzymes at specific times and in specific tissues. Determining how these genes evolved such common *cis*-regulatory elements will require further investigation.

The use of two T2T gapless genomes allowed us to gain insights into complex genomic regions, such as centromeres, telomeres and filled gaps. Centromeres have been extensively studied in the T2T genomes of human[13], *Arabidopsis*[16] and rice[20], all of which contain abundant high-copy tandem repeats. However, the centromeres of *Capsicum* lacked such satellites but were enriched with *Gypsy*-LTR, especially CRM retrotransposons. We also found that this pattern was common in *C. baccatum*, *C. chinense* and *C. pubescens* based on recently reported high-quality genomes[11], although the composition of CRM retrotransposons varied among different *Capsicum* species. The potato genome also showed the enrichment of CRM in centromeres[21], but no CRM was identified in the whole genome of *P. pruinosa* (1.38 Gb, contig N50: 82.2 Mb)[42], suggesting that Solanaceae centromeres evolved rapidly and were diverse among species. Plant centromeres are rapidly diversifying due to cycles of transposon invasions even within species, as shown by a recent study on 346 *Arabidopsis* centromeres[34] that demonstrated the so-called centromere paradox. With more complete *Capsicum* genomes available, comparing centromeres among different *Capsicum* accessions or species to understand the evolution of centromeres during speciation and domestication will be interesting.

In summary, in this study, we have produced so far the largest complete plant genome assembly (*C. annuum*) and two T2T gap-free genomes for *Capsicum*. Phylogenomics and multiomics based on the *Capsicum* T2T genomes unraveled the evolutionary mechanisms underlying the unique and tissue-specific accumulation of capsaicinoids in pepper fruits. These T2T genomic resources are an important milestone in crop genomic research and will accelerate pepper research and promote precise improvement.

## Methods
### Plant materials and sequencing
*C. annuum* double haploid line 'G1-36576' and *C. rhomboideum* wild accession 'PI 645680' plants were grown in regular azalea pots filled with a combination of potting mix, clay and vermiculite in the greenhouse at the Peking University Institute of Advanced Agricultural Sciences, Weifang (36° 42′ N and 119° 10′ E), Shandong Province, China in the summer and autumn of 2022. Fresh leaves of four-week-old *C. annuum* and *C. rhomboideum* were harvested and subjected to DNA extraction and sequencing. Leaf, root, stem and flower tissues at two days post anthesis, and fruits (separated into pericarp, placenta and seeds) at 21 days post anthesis were collected from *C. annuum* and *C. rhomboideum* for RNA extraction and sequencing. Leaf, placenta, pericarp and seeds tissues at 21 days post anthesis were collected from *C. annuum* and used for ATAC-seq. Leaf and whole fruit tissues at 21 days post anthesis were collected from *C. annuum* and used for bisulfite sequencing.

### DNA and RNA isolation
Isolation of high-molecular-weight (HMW) genomic DNA was conducted using the cetyltrimethylammonium bromide (CTAB) method. Briefly, 10 μg of clean and fresh leaves were ground in liquid nitrogen and then subjected to DNA extraction. The quality of the DNA was checked using a Qubit instrument (Thermo Fisher Inc.) and a pulse field gel electrophoresis apparatus (Bio-Rad) according to the manufacturer's instructions. Total RNA was isolated using TRIzol RNA extraction reagent (15596018CN, Thermo Fisher Inc.) according to the manufacturer's instructions. The extracted RNA was assessed using the RNA Nano 6000 Assay Kit of the Bioanalyzer 2100 system (5067-1511, Agilent Technologies, CA) according to the manufacturer's instructions. RNA samples with an RNA integrity number (RIN) > 6.0 were subjected to downstream library construction for RNA sequencing.

### Genome sequencing
Illumina paired-end sequencing library was prepared using the NEBNext® Ultra™ DNA Library Prep Kit for Illumina (E7645L, NEB, USA) according to the manufacturer's standard protocol. Briefly, 5 μg of the HMW DNA sample was fragmented by sonication to a size of 350 bp. DNA fragments were then end-polished, A-tailed, and ligated with full-length Illumina sequencing adapters. A total of 300.8 Gb (~100 × genome coverage) of 150 bp paired-end reads were produced using the Illumina NovaSeq 6000 platform by Novogene Biotechnologies, Inc. (Tianjin, China). The clean data were used for the genome survey, genome assembly polishing and assembly evaluation. To generate PacBio HiFi long reads, a total of 15 μg of HMW DNA was sheared by gTUBEs (Covaris, MA, USA) and used to construct a standard PacBio SMRTbell library via the PacBio SMRTbell Express Template Prep Kit 2.0 (PacBio, CA, USA). The resultant library was separated on Blue-Pippin (Sage Science, MA, USA) with a 15 kb cutoff to remove short DNA fragments. Then, 356.3 Gb of HiFi consensus reads with an N50 length of 18.3 kb were generated using a PacBio Sequel II system at Novogene Biotechnologies, Inc. (Tianjin, China). To generate Oxford Nanopore ultralong reads, long DNA fragments were size-selected and processed using the Ligation Sequencing SQK-LSK109 Kit (Oxford Nanopore Technologies, Oxford, UK) according to the manufacturer's instructions. Briefly, the DNA ends were formalin-fixed and paraffin embedded (FFPE) and end-prepped/dA-tailed using the NEBNext End Repair/dA-tailing module (New England Biolabs, UK). Then, sequencing adapters were ligated onto the prepared ends using the NEBNext Quick Ligation module (New England Biolabs, UK). The final DNA library was sequenced using a GridION X5/PromethION sequencer (Oxford Nanopore Technologies, Oxford, UK) via the Single-Molecule Sequencing Platform at the Peking University Institute of Advanced Agricultural Sciences (Weifang, China). A total of 261.5 Gb of ultralong reads were generated, with read length N50s of 100.3 kb. The Hi-C library was prepared from cross-linked chromatin of pepper leaves using a standard Hi-C protocol[43]. Then, the library was sequenced using an Illumina NovaSeq 6000 instrument to obtain 2 × 150 bp paired-end reads at Novogene Biotechnologies, Inc. (Tianjin, China). A

total of 348.2 Gb of Hi-C data with ~112× coverage was generated and classified as valid or invalid using HiC-Pro v3.1.0[44]; only valid interactions were retained for subsequent analysis.

## Genome assembly

Step 1: The genome size and heterozygosity of DH line 'G1-36576' were estimated using Illumina data by Jellyfish v2.3.0 (*k*-mer size = 19)[45] and GenomeScope v1.0 (max *k*-mer coverage = 1,000,000)[46]. The estimated genome size was 3.19 Gb and the heterozygosity rate was 0.207%.

Step 2: For the PacBio assembly, HiFi reads were assembled using hifiasm (v0.16.1)[24] with the default parameters. The ONT assembly was conducted using NextDenovo (v2.5.0)[25] and polished using NextPolish (v1.4.0)[47] with parameters of 'hifi_options = -min_read_len 1k -max_depth 100' and 'sgs_options = -max_depth 100 -bwa'. Then we aligned the contigs to the reference genomes of *C. annuum* chloroplasts (GenBank accession NC_018552.1) and mitochondrial (GenBank accession NC_024624.1) with Minimap2 (v2.24)[48]. Contigs with at least 50% of their bases covered by chloroplast or mitochondria genome sequences were removed from the assembly.

Step 3: Quickmerge[49] was used to joint contigs in the HiFi assembly (as the query) using the contigs from the ONT assembly (as the reference), which created a HiFi and ONT hybrid assembly. As ONT long reads are usually error-prone, thus we aligned the HiFi contigs to the merged assembly, and replaced the ONT originating sequences with corresponding HiFi contigs. After this step, ten nearly complete chromosome-level contigs were obtained, while two contigs corresponding to Chr04 and Chr08 possessed only a single telomere. Then Hi-C sequencing data were used to anchor all contigs via the Juicer (v1.5)[50], 3D-DNA (v180419)[51] and Juicebox (v1.11.08)[26] pipelines. For assembly validation, the contigs were manually checked and orientation tuned, and any misassembly was adjusted within Juicebox[26].

Step 4: The rDNA arrays on the acrocentric Chr08 were long tandem repeats of the 45S unit (18S-5.8S-25S rDNA). To assemble the 45S rDNA arrays, we first estimated the number of rDNA copies. We used Barrnap v0.9 (https://github.com/tseemann/barrnap) to predict the location of rDNA in the HiFi reads and extracted the 45S rDNA-containing HiFi reads. The copy number was estimated to be ~60,000/42 = 1,428 based on the 19-mers of 45S rDNA-containing HiFi reads (>20 kb, 42×depth). Two main types of repeat unit with different length were identified, Type A (8351–8377 bp) and Type B (8498–8506 bp), each accounting for 70% and 30%, respectively, of the total 45 S rDNA arrays. To assemble the rDNA tandem arrays, we utilized the centroFlye HOR pipeline[52] as a reference. Due to the high similarities between rDNA units and the error-prone property of ONT long reads, we failed to assemble the NOR regions using ONT data. The 45S rDNA contained ONT long reads were used to extract prefix reads that contained telomeric repeats, internal reads that contained two types of rDNA at both ends and suffix reads that contained non-rDNA sequences. Then we assembled the 45S rDNA containing HiFi reads using hifiasm[24] to generate a draft rDNA assembly (78 contigs, N50 of 445.3 kb, and sum of 15.5 Mb). By combining the extracted ONT reads and the assembled HiFi contigs, we identified rare 19-mers and connected the sequences with the same unique 19-mers. We then used Hi-C data to anchor these sequences and filled the gaps by mapping the HiFi reads to the rDNA assembly using Winnowmap2 (v2.03, k = 19, -x asm5)[53]. We finally obtained 12.66 Mb rDNA arrays with 1,506 rDNA copies, and added this sequence to the contig of Chr08.

Step 5: We extracted ultralong ONT reads (>200 kb) with at least ten copies of the telomeric repeat motif 'TTTAGGG' or 'TTCAGGG' variant, and aligned these reads to the above genome assembly using Winnowmap2 (v2.03, k = 19, -ax map-ont)[53]. Using these alignment coordinates, the overhang sequences of telomere-containing reads were manually patched to each telomere. Telomeres were then manually confirmed to be structurally valid. Finally, we obtained a T2T

genome assembly of *C. annuum* accession G1-36576 and named it CaT2T. Similarly, the *C. rhomboideum* genome was assembled using the same strategy described above. The final assembly was named CrT2T.

## Genome quality assessment

To assess the quality of the genome assembly, we first compared the genomic alignment dot plots of the CaT2T and Ca59 assemblies using Minimap2[51] and D-GENIES[54]. For mapping statistics, the NGS short reads were mapped using BWA (v0.7.17)[55], and the HiFi and ONT long reads were mapped using Minimap2[48]. Then SAMtools (v1.10)[56] was used to determine the mapping rates and coverage depth. The Ca59 gap regions that were resolved in the CaT2T genome were manually checked in IGV (v2.12.3)[57]. To assess genome completeness, we applied BUSCO (v5.4.3)[58] for ortholog detection using the solanales_odb10 database (n = 5,950). The quality value (QV) was estimated using Merqury (v1.3)[59] from HiFi reads. The telomere sequences were identified using Tandem Repeat Finder (TRF, v4.09.1)[60] with the parameters of '2 7 7 80 10 80 2000 -d -l 16'. The resulting '.dat file' was transformed into a GFF3 file, which was subsequently used to identify seven base telomeric repeats.

## Repeat annotation and TE analysis

We used the universal Repbase database and a species specific de novo repeat library constructed by RepeatModeler (https://github.com/Dfam-consortium/RepeatModeler) to annotate the DNA sequences of the two *Capsicum* species. The repetitive elements in the genome were then annotated and masked by RepeatMasker (v4.1.2)[61] using the following parameters: '-xsmall -s -no_id -cutoff 255 -frag 20000 -e ncbi'. To achieve large-scale accurate discovery of LTR retrotransposons, we applied LTR_Finder (v1.2)[62], LTRharvest (v1.6.2)[63], and LTR_retriever (v2.9.0)[64] to identify LTR elements. We identified 7383 and 9579 intact LTR-RT candidates in CaT2T and CrT2T, respectively, which were used as inputs for the TEtranscripts analysis[65]. TEsorter (v1.3)[66] was subsequently implemented using HMM profiles obtained from the TE protein domain database REXd-plant. The TE sequences were first translated in all six frames and the translated sequences were then searched against the database. Hits with coverage lower than 20% or an E-value higher than 1e-3 were discarded. For the classification of LTR-RTs, intact elements were identified and classified based on the presence and order of five conserved domains, including capsid protein (GAG), aspartic proteinase (AP), integrase (INT), reverse transcriptase (RT), and RNase H (RH). After filtering the conserved domains, the number of LTR-RTs decreased to 5202 and 6834 in CaT2T and ChT2T, respectively. Using TEsorter (v1.3)[66], the Ty1-*Copia* elements were classified into several clades, including *Ale, Alesia, Angela, Bianca, Ikeros, Ivana, SIRE, TAR*, and *Tork*; while the Ty3-*Gypsy* elements were classified into clades of *Athila, CRM, Galadriel, Ogre, Reina*, and *Tekay*. The ratios of solo LTRs to intact LTRs in each LTR family were calculated using the script of 'solo_intact_ratio.pl' in LTR_retriever software. The insertion times of the intact LTR retrotransposons were calculated using LTR_retriever according to the formula:

$$T = K/2r \tag{1}$$

where K is the divergence between the two LTRs and r is the rate of nucleotide substitution. We employed an average substitution rate of (r) $7 \times 10^{-9}$ to estimate the insertion times of LTR-RTs.

## Genome annotation

Gene model prediction combined with the following three aspects of evidence: (a) ab initio prediction, (b) homologous protein, and (c) RNA-seq evidence, was conducted using MAKER (v2.31.11)[67] pipeline in two successive rounds. In the first round, short-read and full-length RNA-seq evidence and homology proteins were provided. The protein

sequences used for homology-based prediction were from *A. thaliana*[16], *C. annuum*[11], *S. tuberosum*[21], and universal Swiss-Prot proteins. To implement the MAKER pipeline, short-read RNA-Seq data were assembled into a transcriptome using StringTie (v2.2.1)[68]. The PacBio long-read transcriptome data were processed using the SMRT Analysis software Isoseq3 (https://github.com/PacificBiosciences/IsoSeq). BLAST was employed to align transcripts and proteins to the soft-masked genome via MAKER, then Exonerate (v2.2.0)[69] was used to polish the BLAST hits and thereby accurately annotate the coding regions. The parameters of est2genome and protein2genome were set to 1, so that MAKER predicted gene models based only on the provided transcripts and proteins. Then a subset of MAKER gene models with AED[70] scores <0.25 was used to train SNAP[71] for three rounds. The GeneMark-ET and Augustus models were trained using the BRAKER (v2.1.6)[72] pipeline. Briefly, the same data were aligned to the soft-masked genome using Exonerate[69] and HISAT2 (v2.1.0)[73]. Then GeneMark-ET[74] was trained on the predicted gene structures, and the resulting ~6,000 good gene models were used for training AUGUSTUS (v3.2.3)[75]. In the second round, each set of gene predictions in round one was passed to MAKER through the model_gff option and the evidence alignment options were turned off. The trained SNAP, GeneMark-ET and AUGUSTUS models were also integrated into MAKER to predict more credible genes. Finally, the unsupported gene models were filtered (keep_preds=0), and the highest-ranking gene sets with AED scores <0.5 were retained.

To compare the previously published genome annotation of *C. annuum*[4,11] with our CaT2T genome annotation, we also performed Liftoff (v1.6.3)[76] to annotate protein-coding genes of the CaT2T assembly based on a reference with the parameters of "-flank 0.1 -sc 0.99 --copies". Then Gffread (v0.12.7)[77] was used to filter transcripts without normal open reading frames. Gene models were finally manually checked and corrected in IGV-GSAman (v0.6.76) (https://gitee.com/CJchen/IGV-sRNA) with the support of mapped RNA-seq reads and previous annotations[4,11].

### Synteny and phylogenome analysis
Nonredundant protein sequences from 16 species were prepared for ortholog analyses (Supplementary Table 10). Orthologs and orthogroups were then inferred using OrthoFinder (v2.5.4)[35] with the default settings and '-M msa' activation. The longest predicted protein of each individual gene was used as the representative input for the OrthoFinder analysis. TrimAl (v1.4.12)[78] was used to remove poorly aligned regions of protein multiple sequence alignments. RAxML (v8.2.12)[79] was used to construct maximum likelihood phylogenetic trees using the GAMMAJTT model, with rice as an outgroup. TimeTree (www.timetree.org) is a public database containing divergence time estimates from various publications, along with their own estimations. These estimates, ignoring the outliers, were used for selecting the range of lower and upper uniform calibration priors. The calibration values were chosen as 1.1–1.6, 109.2–123.5, and <200 for the most common ancestor of the 13 species belonging to Solanum, dicotyledons, and all plants, respectively. The CodeML and MCMCTree programs in PAML (v4.9)[80] were used to analyze amino acid substitution models and estimate divergence times. CAFE5[81] was then used to infer the gene gain and loss rates in each genome. The orthogroups generated by OrthoFinder were regarded as distinct gene families and provided as inputs for CAFE5 analysis. The identified genes were subjected to Gene Ontology (GO) and Kyoto Encyclopedia of Genes and Genomes (KEGG) enrichment analyses, and the *p* value indicating significant enrichment was set as 0.05. The syntenic analysis was performed by JCVI (v1.1.19)[82]. We identified synteny blocks by performing an all-against-all LAST search and chaining the hits with a distance cutoff of 20 genes. Additionally, we required each synteny block to have at least five gene pairs. A dot plot of the major CBGs is shown in Gepard[83]. The *Ks* values of *C. annuum* syntenic block genes were calculated using ParaAT (v2.0)[84].

### CENH3 ChIP-seq
An antigen with a full peptide sequence corresponding to *C. annuum* CENH3 was used to produce *C. annuum* anti-CENH3 antibodies in rabbits. The preparation and affinity purification of antisera were conducted by the AtaGenix (Wuhan, China). For the ChIP experiment, pepper seedlings were fixed with 1% formaldehyde solution in MS buffer (10 mM potassium phosphate, pH 7.0; 50 mM NaCl) at room temperature for 15 min under vacuum. After fixation, the seedlings were incubated at room temperature for 5 min under vacuum with 0.15 M glycine. Approximately 1 g of fixed tissue was homogenized with liquid nitrogen, and the nuclei were purified, resuspended in 1 ml of cell lysis buffer, incubated for 10 min on ice, and spun at 1500 rpm (RC-3B, 600 × *g*) for 5 min (cell lysis buffer: 10 mM Tris, 10 mM NaCl, 0.2% NP-40 [pH 8.0], 1× protease inhibitors). The cell lysate was further resuspended in 1 ml of nuclear lysis buffer for 10 min on ice (nuclear lysis buffer: 50 mM Tris, 10 mM EDTA, 1% SDS, 1× protease inhibitors) to isolate the nuclei. The resuspended chromatin solution was sonicated five times for 15 s each at ~10% power (setting 2.5 on the sonicator, Sanyo Soniprep 150). The volume of the chromatin sample was measured, and then, ChIP dilution buffer was added to 1 ml of chromatin with 2.5 μg of anti-H3K4me3, and the samples was incubated for 12 h at 4 °C. Then, 50 μl of protein A/G beads were added, and the sample was incubated for 4 h at 4 °C. The beads were washed twice with each of the following buffers: wash buffer A (50 mM HEPES-KOH pH 7.5, 140 mM NaCl, 1 mM EDTA pH 8.0, 0.1% Na-deoxycholate, 1% Triton X-100, 0.1% SDS), wash buffer B (50 mM HEPES-KOH pH 7.9, 500 mM NaCl, 1 mM EDTA pH 8.0, 0.1% Na-deoxycholate, 1% Triton X-100, 0.1% SDS), wash buffer C (20 mM Tris-HCl pH 8.0, 250 mM LiCl, 1 mM EDTA pH 8.0, 0.5% Na-deoxycholate, 0.5% IGEPAL C-630, 0.1% SDS), wash buffer D (TE with 0.2% Triton X-100), and TE buffer. To purify the eluted DNA, 200 μl of TE was added, and the RNA was degraded by the addition of 2.5 μl of 33 mg/mL RNase A (Sigma, R4642) and incubation at 37 °C for 2 h. The DNA was then resuspended in 50 μl of TE and amplified with the VAHTS® Universal DNA Library Prep Kit for Illumina V3 (Vazyme ND607). Amplified ChIP libraries were sequenced on the Illumina NovaSeq 6000 platform.

### Epigenomic sequencing and data analysis
Hi-C data were generated from leaf tissue as mentioned above and processed using HiC-Pro (v3.1.0)[44] and Juicertools (v1.22.01)[53] to generate 10 kb, 15 kb, 20 kb, 25 kb, 40 kb, 100 kb and 500 kb contact maps. A/B compartments were identified by R (v4.2.0) script using HiTC (v1.42.0)[85] and Cworld-dekker (v0.0.1) (https://github.com/dekkerlab/cworld-dekker) in a 100 kb iced contact matrix. The chromatin accessibility of *C. annuum* was profiled using an ATAC-seq construction kit (Vazyme Ltd. Nanjing China) according to the manufacturer's protocol. The quality of the constructed libraries was assessed using a qubit followed by an Agilent Bioanalyzer 2100 for fragment analysis. The libraries were sequenced using an Illumina NovoSeq 6000 platform at Novogene, Inc. (Tianjin, China). Three biological replicates were generated for each plant tissue and analyzed using the same computational methods. The ATAC-seq data were analyzed using an in-house computational pipeline. Basically, the clean ATAC-seq reads were mapped to the CaT2T reference genome using BWA-MEM (v2.2.1)[86] with default parameters. The alignment files (.bam) were used to call peaks by MACS2 (v2.2.7.1)[87]. Histone modification ChIP-seq data were downloaded from the public CNGBdb database with accession number CNP0001129. Whole-genome bisulfite sequencing was conducted on leaf and whole fruit tissues from *C. annuum*. ChIP-seq mapping and peak calling were performed with commands using Bowtie2 (v2.5.1)[88], SAMtools (v1.10)[56], and MACS2 (v2.2.7.1)[87]. The visualization of centromeric repeats was accomplished using StainedGlass[89]. The DNA methylation level was estimated using Bismark (v0.24.0)[90] after mapping whole-genome bisulfite sequencing (WGBS) data to the reference genome using BWA-MEM (v2.2.1)[86].

## Transcriptome sequencing and analysis

Total RNA was extracted from seven tissues, including leaf, flower, placenta, root, stem, seeds and pericarp. The mRNA was then subjected to transcriptome sequencing library construction using an Illumina True-seq transcriptome kit (Illumina, CA). The libraries were then sequenced using the Illumina NovaSeq 6000 platform at Biomarker Technologies Corporation (QingDao, China) to generate 150 bp paired-end reads. For full-length transcriptome sequencing, approximately 5 μg of mRNA was reverse-transcribed into full-length cDNA with a SMARTer™ PCR cDNA Synthesis Kit (Clontech, CA, USA), and the cDNA was further amplified by PCR. End repair was conducted on amplified cDNAs, followed by SMRTbell adapter ligation. The ligation products were further treated by exonuclease to degrade the failed products before the Iso-Seq library was sequenced using a PacBio Sequal IIe instrument at Biomarker Technologies Corporation (QingDao, China). Full-length transcripts were assembled across tissues using the SMRTlink pipeline and used for guiding gene annotation. We quantified gene expression levels using kallisto (v0.48.0)[91]. Counts for mapped reads were normalized by transcripts per million (TPM). Read alignment was performed using HISAT2 (v2.1.0)[73]. To visualize the expression patterns of the genes of interest among the samples, heatmaps were generated using the R package.

## Reporting summary

Further information on research design is available in the Nature Portfolio Reporting Summary linked to this article.

## Data availability

The raw sequencing data (PacBio HiFi, ONT, Illumina paired-end, Hi-C and RNA-seq) and genome assembly generated in this study have been deposited in the National Center for Biotechnology Information (NCBI) under accession code PRJNA962192. The genome assembly and annotation files are available at *Capsicum* Genome Database [http://www.pepperbase.site/node/3] of Peking University Institute of Advanced Agricultural Sciences. Source data are provided with this paper.

## Code availability

Custom scripts and codes used in this study are available at GitHub [https://github.com/Weikai-47/Pepper_T2T] and Zenodo [https://doi.org/10.5281/zenodo.11078975][92].

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

## Acknowledgements
We would like to thank the Bioinformatics Platform at Peking University Institute of Advanced Agricultural Sciences for providing the high-performance computing resources. We would like to thank Zenghui Chen for technical assistance, and East-West Seed Group for kindly providing the double haploid pepper seeds. This work was supported by the Key R&D Program of Shandong Province (Grant No. ZR202211070163, L.G.), Taishan Scholars Program (L.G.) and Natural Science Foundation for Distinguished Young Scholars (Grant No. ZR2023JQ010, L.G.) of Shandong Province, Xizang Autonomous Region of Lhasa City Science and Technology Project (Grant Nos. LSKJ202418; LSKJ202310, Z.Z.).

## Author contributions
L.G., H.H., X.W.D. and X.Z. conceived and designed the project. Y.M., J.B., B.L., J.L., L.W. and Z.Z. maintained the plant materials and coordinated sequencing data generation. W.C. and X.F.W. assembled the genomes and performed genome validations and annotations. X.F.W., X.R.W., W.C., J.S., S.Y., D.H.A., M.Y., and K.W. conducted bioinformatic analysis and prepared figures and tables. D.M. conducted epigenome sequencing. S.C. and T.M. constructed the genome database. L.Z., R.C. and J.J. assisted in the result interpretation and discussion. L.G., W.C., L.Z. and D.H.A. wrote the manuscript. L.G., X.W.D., X.Z. and H.H. revised the manuscript. All authors read and approved the final version of the manuscript.

## Competing interests
The authors declare no competing interests.
