## [Peer Review File · Nature Communications]

Two telomere-to-telomere gapless genomes reveal insights into Capsicum evolution and capsaicinoid biosynthesisReviewers' Comments:

Reviewer #1:

Remarks to the Author:

This is a nice manuscript that describes the assembly of two very high quality telomere-to-telomere *Capsicum* genomes in great detail as well as the analyses of centromere composition and capsaicinoid biosynthesis genes. A putative placenta-expressed gene region was identified using chromatin accessibility data. This work includes the first wild pepper genome and a fair amount of epigenomic data.

Centromeric Retrotransposons of Maize (CRM) were found in the centromeres of both genomes, but the biological significance of this is not clear. Because retrotransposons are featured at great length in the centromere discussion, it is essential to provide a sequence file or sequence coordinates (gff) for each genome to allow verification. This could be done for all, or only the most appropriate of the three datasets (17,503 and 22,148 OR 7,383 and 9,579 OR 5,202 and 6,834 - all data sets listed on page 23) retrotransposons shown in Figure 2D.

A machine learning model was used to model capsaicin production relative to copy number variations or relevant genes and showed promise, although the authors report that due to the small training set it revealed primarily already known relationships. This is a promising approach.

The two high-quality genomes and containing many beautiful figures reported in this publication will be of high public interest.

The manuscript states that "The genome assembly, annotations and SVM-model for pungency classification are also available in *Capsicum* Genome Database (<http://www.pepperbase.site>) of Peking University Institute of Advanced Agricultural Sciences. (<http://www.pepperbase.site>)" - but I was unable to find them there. The genomes are available without annotation as FASTA files on GenBank, but really should be made available to the public in the form of gff files and/or genome browser.

Line 81 should be "rice" instead of "rich".

Reviewer #2:

Remarks to the Author:

Chen et al. report T2T genome assemblies of a cultivated pepper (*Capsicum annuum*) and a distantly related wild species (*C. rhomboideum*), adding important resources to the pepper genomics. The authors then performed a series of analyses to obtain insights into *Capsicum* evolution and the regulation of capsaicinoid biosynthesis, and used a machine-learning model (support vector machine; SVM) to predict pungency levels with copy number variants (CNVs).

Major:

Line 114-115: the total length of the filled gaps should be provided. It is interesting that after gap filling, the sizes of the final T2T genomes are even smaller than the HiFi assemblies (16.8 Mb and 17.7 Mb less for CaT2T and CrT2T, respectively; Table S2). What sequences were discarded, and why?

The authors predicted ~45-48K protein-coding genes in the two genomes (Line 147), much more than those predicted in most previously published pepper genomes and other Solanaceae genomes (~35K). In addition, BUSCO evaluation (v5.4.7 with *embryophyta_odb10*) of the predicted genes indicates that only 79.9% and 86.9% conserved single-copy orthologs were completely annotated in the CaT2T and CrT2T genomes, respectively. These suggest both a high false positive rate and a high false negative

rate of gene predictions in these two assemblies, which would substantially complicate the downstream analyses.

It is very surprising that in the species tree (Fig. 4A), *Solanum tuberosum* (potato) clustered more closely with *S. melongena* (eggplant) than with *S. lycopersicum* (tomato). This is clearly wrong. The authors can check <https://www.nature.com/articles/s41438-020-00391-0/figures/3> and <https://www.nature.com/articles/s42003-021-02152-8/figures/1> and many others. The tree topology for the Solanaceae species is not consistent with many previously reported ones. Therefore, results from this section could be unreliable. In addition, in Fig. 4A, the authors missed labeling a WGT event in *Ipomoea* (<https://www.nature.com/articles/s41467-018-06983-8>), and a WGD in tea (<https://www.nature.com/articles/s41438-021-00613-z>). Furthermore, the authors should use the recently reported higher quality and better annotated genome of *C. baccatum*, and also include the newly generated *C. pubescens* genome in the analysis. Nonetheless, this entire section on capsaicinoid biosynthesis pathway genes (expressed higher in pungent species and SVs of these genes present between and within pungent and non-pungent species) is quite descriptive and does not provide much novel insight.

Line 317-341: The data described in this section were not comprehensively analyzed and integrated. No description on the results of bisulfite sequencing. The authors mentioned that placenta-specific OCRs were detected for CS (Line 324-325); however, according to Fig. 4F, OCRs were also detected in pulp and seed for CS-1. In addition, the insights obtained from this section seem limited.

Line 351-358: In this section, the authors were attempting to demonstrate that a T2T genome can improve variant calling. First, I found this entire section quite boring as it is obvious that a more complete genome would improve variant detection. Second, while the T2T genome can improve variant calling, the improvement is minor (<1%; Line 359 and 362). Therefore, their conclusion (Line 38-40 and 467-468) is overstated ("substantially", "significantly"). Third, the authors derived their conclusions mainly based on the numbers of called variants. The more does not mean the better in variant calling. A deeper analysis is needed to reach a robust conclusion. In addition, if the authors want to emphasize the importance of "T2T", a more logic comparison would be variants called using the T2T genome and those with the HiFi-based assembly (the assembly prior to "T2T"). Nonetheless, I think this section does not provide much novel information and can be removed.

Line 369-395: It's widely acknowledged that identification of CNVs based on short-read data presents a high false positive rate. To ensure the reliability of the identified CNVs, evaluation of the CNV accuracy is necessary. In addition, the list of these CNVs should be provided in a supplementary table. It appears challenging to see from Fig. 6A that "For cultivars with intact CS gene, capsaicinoid levels appeared to be affected by CNVs of other CBGs" (Line 378). Statistical analyses are needed here.

Minor:

Line 64 and 118: Genome sequences of wild peppers are available, e.g., *C. annum* var. *glabriusculum* (<https://pnas.org/doi/full/10.1073/pnas.1400975111>).

Please add the accession name for *C. rhomboideum* that was used for genome assembly.

Fig. 1A panel b: the beginning of Chr08 of CaT2T has a very high GC content compared to other genomic regions. Have the authors checked what are the possible reasons that cause this?

Table S4: why LAI of CaT2T was much lower than CrT2T. The authors should check all other published pepper genomes to make sure that cultivated peppers indeed have a low LAI.

Line 280-282: As the authors stated, this could be "a result of gene loss", then how this can highlight the importance of T2T assembly? Much deeper analysis is needed to reach such conclusion (e.g., confirm the genes were present in *C. baccatum* but not assembled).

Line 287-289: Please be careful to describe the results here. From Fig. 4D, it is clear that NOT all CBGs displayed high expression levels in pungent Capsicum species.

Line 295-296: The authors mentioned just above that SVs are present in CDS, making the statement here invalid.

CNVs were identified for 75 of 124 CBGs (Line 375). It is confusing that how "the SVM model ranked 124 CBGs" (Line 387) as CNVs were used in the model while 49 CBGs do not harbor CNVs?

Line 411-412: Only four Capsicum genomes were used (Fig. 4A), instead of six.

Reviewer #3:

Remarks to the Author:

This study assembled two T2T gapless Capsicum genomes for the pungent pepper *C. annuum* and its non-pungent wild relative *C. rhomboideum*. The authors examined distinct structural, epigenetic, and transcriptional features in their centromeres to delineate the evolution of fruit pungency in chili peppers. Specific chromatin regions associated with tissue-specific gene regulation were identified. The complete *C. annuum* genome improved variant detection and was used to create a machine learning model to predict pungency for genomic selection. However, I would like to suggest a substantial revision for this manuscript.

1. Genome validation. Although the authors reported T2T genome of two pepper species, more data to accept that those are really completed "Finished" T2T genomes. The authors should evaluate genome quality based on reliable evaluation process. The genome validation by the earth genome project can be helpful (<https://www.earthbiogenome.org/assembly-standards>). The authors need to confirm that those T2T genomes satisfy conditions for "Finished" suggested by the earth genome project
2. In addition, the previously reported genome assemblies were used to highlight the genome quality of the CaT2T (*C. annuum*) assembly, capsicinoid gene analyses and variation detections. However, the latest genome version was recently reported and thus should be used in the analysis. *C. annuum* and *C. baccatum* genome assemblies were recently released in 2023 (Nat. Comm). Genome comparison, analyses for pungency-related genes (genes in *C. baccatum*) and highlighting variation detection parts should be updated after including additionally those most recently published genomes.
3. Total number of genes are over 45000 in both pepper genomes. These are too many comparing to previous pepper genomes and other Solanaceae genomes. Sustainable validation of gene annotation should be performed in addition to BUSCO. BUSCO analyses can only prove partial accuracy. In particular, the authors should evaluate and analyze over 10000 genes specifically in two pepper genomes in this study.
4. The authors used Hi-C data of CA59 to analyze TADs in CaT2T. However, the epigenomic structures such as TADs are different between i) accessions including between same accessions. I cannot agree that the authors have concluded their findings based on TAD analyses using CA59 for different accession as reference genome. To generalize, the authors should use Hi-C data for multiple accessions in addition to CA59 and should conclude their results.
5. Genome annotation predicted that the CaT2T and CrT2T centromeres contain 252 and 386 genes, respectively. The authors identified the divergence of centromere genes between two species and suggested their potential function based on the GO analysis. Although there are many centromere-encoded genes in *C. rhomboideum*, only RNA-seq data for *C. annuum* was used to describe transcriptional activity. Why were the centromere-encoded genes in CrT2T not identified?
6. In *C. annuum*, centromere-located CRMs showed lower CHG methylation of gene-body, suggesting transcriptional activity of CRMs. Have you examined it for the other gypsy subfamilies such as athila and tekay?
7. Sequence alignment of capsicinoid biosynthesis genes (CBGs), such as ACL, BCAT, CCoAMT, FatA,

and KasI, showed that sequence variations were observed in Capsicum species. Moreover, it was identified that all CBGs were highly expressed in fruits of pungent Capsicum species, while hardly expressed in *C. rhomboideum* and *Physalis*. It is also well known that the biosynthesis of capsaicin was occurred in the fruit placenta tissue of pepper. Have these sequence variations and expressions for each tissue been experimentally validated? As I mentioned above, *C. baccatum* genome was recently published and the authors need to check CS genes are really low copy in the high-quality *C. baccatum* genome.

8. This study suggests that MYB31 and MYB48 regulate several key genes responsible for capsaicin biosynthesis in pepper and DNA methylation in specific regions of them was detected using bisulfite sequencing. It is known that the expression level of not only transcription factors but also structural genes, which were related to the capsaicin biosynthetic process, can be regulated by the hypomethylation or hypermethylation at the gene level. Have authors investigated DNA methylation and expression features of genes that might be co-expressed with MYB31 and MYB48 in pungent and non-pungent peppers? This analysis can help to understand the model for the Capsicum-unique and tissue-specific regulation of the capsaicin biosynthesis genes in the hot pepper.

Minor

- Table S1 genome size estimation of *C. rhomboideum* is calculated using 1.71Gb.

Reviewer #4:

Remarks to the Author:

This review focuses on the machine learning aspects of the genomic prediction in the paper.

To start, some of the comments here are only possible after reviewing the github code (notably `Gridsearch.py`). Many of the details should be presented in the paper or supplementary material (e.g., hyperparameter ranges, etc.) The optimal hyperparameters from the grid search on the SVM must be presented in the paper (presumably they are RBF kernel with $C = 0.01$ and gamma the default in `scikit-learn`).

The use of a binary classification framework rather than doing regression is not motivated, and is an unnecessary simplification. If there is a motivation for the binary classification in real-world terms? The authors should show the results of using an SVR to do prediction of the raw capsaicinoids content using the same hyperparameters as the classification problem, or preferably a new grid search for the regression problem. If this is not done, the authors should at a very minimum provide motivation for this choice of classification (presumably, because it is easier and gives better results).

The authors acknowledge the limitations of the training set size, but also employ 30-fold CV with their training. With a training set of 311 elements, this gives a test set of approximately ten in each fold. This is too small, creating the possibility of larger variation between folds. I would suggest a fold number in the range of 5-10 to verify the same results hold (Strangely, the github code reveals a cv parameter of 80. Surely this must not be the value that was used.) If the authors feel that 30-fold CV is required, which would be very unusual, there should be some citation to the literature as to the requirements, especially in the context of the small training set. In the absence of this, redoing the experiments with a CV with 5-10 folds would be necessary.

The grid search for RBF without searching for optimal values of gamma is unusual. The authors should consider this hyperparameter in their grid search, or otherwise validate why this was not done. Code for the Decision tree, Random forest or KNN is not included in github and insufficiently described in the paper. The range of hyperparameters that were considered for each of these models should at a minimum be described somewhere.

Reviewer #1 (Remarks to the Author):

This is a nice manuscript that describes the assembly of two very high quality telomere-to-telomere Capsicum genomes in great detail as well as the analyses of centromere composition and capsaicinoid biosynthesis genes. A putative placenta-expressed gene region was identified using chromatin accessibility data. This work includes the first wild pepper genome and a fair amount of epigenomic data.

RESPONSE: Thank you very much for your valuable comments and suggestions helping us to improve the manuscript. We have carefully revised and improved the manuscript.

Centromeric Retrotransposons of Maize (CRM) were found in the centromeres of both genomes, but the biological significance of this is not clear. Because retrotransposons are featured at great length in the centromere discussion, it is essential to provide a sequence file or sequence coordinates (gff) for each genome to allow verification. This could be done for all, or only the most appropriate of the three datasets (17,503 and 22,148 OR 7,383 and 9,579 OR 5,202 and 6,834 - all data sets listed on page 23) retrotransposons shown in Figure 2D.

RESPONSE: We appreciate the reviewer's comment and suggestion. CRM like retrotransposons are a major component of chili pepper centromeres as well as several other plants such as maize. Although the biological significance of this remains unclear, our ChIP-seq analysis shows that CENH3 protein frequently binds to CRM repeats, suggesting CRMs are likely critical to normal function of centromeres such as segregation of chromosomes during cell divisions. Regarding the coordinates of the CRMs, we have added a BED file specifying the coordinates of CRM elements in the two pepper assemblies as Supplementary Table 10.

A machine learning model was used to model capsaicin production relative to copy number variations or relevant genes and showed promise, although the authors report that due to the small training set it revealed primarily already known relationships. This is a promising approach.

RESPONSE: Thank you very much for the positive comment!

The two high-quality genomes and containing many beautiful figures reported in this publication will be of high public interest.

RESPONSE: Thanks for your positive comment!

The manuscript states that "The genome assembly, annotations and SVM-model for pungency classification are also available in Capsicum Genome Database (<http://www.pepperbase.site>) of Peking University Institute of Advanced Agricultural Sciences. (<http://www.pepperbase.site>)" - but I was unable to find them there. The genomes are available without annotation as FASTA files on GenBank, but really should be made available to the public in the form of gff files and/or genome browser.

RESPONSE: Thank you very much for the suggestion. We have updated the website of our Capsicum Genome Database with the following link (<http://www.pepperbase.site/node/3>) where the genome annotation GFF files can be downloaded. The SVM model can be accessed in the Github site (https://github.com/Weikai-47/Pepper_T2T). Both T2T genomes are available in NCBI GenBank under the BioProject PRJNA962192.

Line 81 should be "rice" instead of "rich".

RESPONSE: Thank you very much. Revised as suggested.

Reviewer #2 (Remarks to the Author):

*Chen et al. report T2T genome assemblies of a cultivated pepper (*Capsicum annuum*) and a distantly related wild species (*C. rhomboideum*), adding important resources to the pepper genomics. The authors then performed a series of analyses to obtain insights into *Capsicum* evolution and the regulation of capsaicinoid biosynthesis, and used a machine-learning model (support vector machine; SVM) to predict pungency levels with copy number variants (CNVs).*

RESPONSE: Thank you very much for your valuable comments and suggestions helping us to improve the manuscript. We have carefully revised and improved the manuscript.

Major:

Line 114-115: the total length of the filled gaps should be provided. It is interesting that after gap filling, the sizes of the final T2T genomes are even smaller than the HiFi assemblies (16.8 Mb and 17.7 Mb less for CaT2T and CrT2T, respectively; Table S2). What sequences were discarded, and why?

RESPONSE: Thank you very much for the insightful feedback. We revised the manuscript to provide the length of filled gaps in HiFi assemblies in Supplementary Table 3. Take the CaT2T as an example, we checked the 245 contigs in HiFi assembly, which included 22 contigs corresponding to main genomic sequences in 3,085.07 Mb, 210 contigs of 45S rDNA repeats in total length of 22.06 Mb, and 13 contigs of probably collapsed sequences in 12.81 Mb located mainly close to telomeres (<5 Mb). Thus, the discarded sequences are mainly redundant contigs and 45S rDNA contigs in repetitive regions.

The authors predicted ~45-48K protein-coding genes in the two genomes (Line 147), much more than those predicted in most previously published pepper genomes and other Solanaceae genomes (~35K). In addition, BUSCO evaluation (v5.4.7 with embryophyta_odb10) of the predicted genes indicates that only 79.9% and 86.9% conserved single-copy orthologs were completely annotated in the CaT2T and CrT2T genomes, respectively. These suggest both a high false positive rate and a high false negative rate of gene predictions in these two assemblies, which would substantially complicate the downstream analyses.

RESPONSE: We very much appreciate your suggestion. Previously, we annotated the two pepper genomes following the pipeline reported in *C. annuum* cv. 59 (46,160 genes, Nat. Comm, 2022) which yielded a comparable number of genes. In the revised manuscript, we updated the pipeline to find high-confidence gene models that satisfy all three sources of evidence including *ab initio* prediction, the homology proteins and transcriptomic evidence. As a result, the unsupported or low-confidence (missing either of the three evidence) gene models were filtered out, generating new

versions of genome annotation with 34,428 and 33,512 protein-coding genes in CaT2T and CrT2T, respectively. BUSCO evaluation using solanales_odb10 database (n = 5,950) indicated 97.04% and 93.23% single-copy orthologs were completely annotated in the CaT2T and CrT2T. We have updated the **Fig.1, Fig.4 and Fig.5** accordingly and revised the manuscript (Line 154).

It is very surprising that in the species tree (Fig. 4A), Solanum tuberosum (potato) clustered more closely with S. melongena (eggplant) than with S. lycopersicum (tomato). This is clearly wrong. The authors can check <https://www.nature.com/articles/s41438-020-00391-0/figures/3> and <https://www.nature.com/articles/s42003-021-02152-8/figures/1> and many others. The tree topology for the Solanaceae species is not consistent with many previously reported ones. Therefore, results from this section could be unreliable. In addition, in Fig. 4A, the authors missed labeling a WGT event in Ipomoea (<https://www.nature.com/articles/s41467-018-06983-8>), and a WGD in tea (<https://www.nature.com/articles/s41438-021-00613-z>). Furthermore, the authors should use the recently reported higher quality and better annotated genome of C. baccatum, and also include the newly generated C. pubescens genome in the analysis.

RESPONSE: We apologize for the error in the phylogenetic tree. Using the updated genome annotations (see above) of *C. annuum*, *C. rhomboideum*, and downloaded annotations of recently published *C. baccatum* and *C. pubescens* genomes, we have reconducted the phylogenetic analysis using single-copy orthologs identified by *Orthofinder*. The revised phylogenetic tree now correctly places *S. tuberosum* closer to *S. lycopersicum* than to *S. melongena* (revised **Fig. 4A**). In addition, as suggested by the reviewer, we have corrected the inaccurate placement of *Ipomoea* and tea WGD event on the tree and updated the version of pepper genomes in the new species tree (revised **Fig. 4A**).

Nonetheless, this entire section on capsaicinoid biosynthesis pathway genes (expressed higher in pungent species and SVs of these genes present between and within pungent and non-pungent species) is quite descriptive and does not provide much novel insight.

RESPONSE: We believe our analysis does provide novel insight despite the descriptive

writing. First, there are no previous studies dedicated to comparing the capsaicinoid biosynthesis pathways at genomic level among multiple Solanaceae plants, probably due to the poor genomic information of *Capsicum* genus until recently and this study. Therefore, we presented the first comprehensive analysis of the pathway and the biosynthesis genes at the phylogenetic context (including estimated timing for key events) for these plant species, providing insight into the evolution of capsaicins and pungency. Second, by comparing the key biosynthesis gene Capsaicin Synthase (CS) among the close related plant species, we offered mechanistic insight into how the regulation of CS genes went wrong in non-pungent peppers even though they contain homologs of the pepper CS gene, suggesting the role structural variants play in the capsaicin evolution. To the best of our knowledge, no study has reported this type of evolution mechanism before regarding the capsaicin biosynthesis. Therefore, our study represents the first comprehensive genomic studies of capsaicin biosynthesis pathway within Solanaceae plants, offering novel insight into how the pathway likely evolved in these species including chili pepper.

Line 317-341: The data described in this section were not comprehensively analyzed and integrated. No description on the results of bisulfite sequencing. The authors mentioned that placenta-specific OCRs were detected for CS (Line 324-325); however, according to Fig. 4F, OCRs were also detected in pulp and seed for CS-1. In addition, the insights obtained from this section seem limited.

RESPONSE: We apologize for the inaccurate writing and insufficient discussion regarding the epigenetic regulation of CBGs. We have added the discussion on the bisulfite sequencing data in the revised manuscript (Line 339). It is true that CS-1 has OCRs detected in both pulp and seed, different from CS which only has placenta-specific OCR. RNA-seq data suggests both CS genes are only expressed in pulp. Our speculation is that the CS-1 OCR likely allowed certain negative transcriptional regulator(s) to bind and suppress its expression in seed. The significance of this whole section lies in that we combined ATAC-seq, bisulfite sequencing and RNAseq to investigate the epigenetic regulation of capsaicin biosynthesis genes allowing the co-regulation of them in fruit tissue, which has not been shown before.

Accordingly, we have revised the relevant parts of manuscript as the following: “The placenta-specific open chromatin regions (OCRs) with low methylation levels were detected within 2 kb

upstream of *CS-2*, *MYB31* and *MYB48*, while *CS-1* also showed OCRs in pulp and seeds, suggesting the *CS-2* is the real functional gene that contribute to placenta-specific synthesis of capsaicinoids (Fig. 4F; Supplementary Fig. 21).” (Lines 339-342)

Line 351-358: In this section, the authors were attempting to demonstrate that a T2T genome can improve variant calling. First, I found this entire section quite boring as it is obvious that a more complete genome would improve variant detection. Second, while the T2T genome can improve variant calling, the improvement is minor (<1%; Line 359 and 362). Therefore, their conclusion (Line 38-40 and 467-468) is overstated (“substantially”, “significantly”). Third, the authors derived their conclusions mainly based on the numbers of called variants. The more does not mean the better in variant calling. A deeper analysis is needed to reach a robust conclusion. In addition, if the authors want to emphasize the importance of “T2T”, a more logic comparison would be variants called using the T2T genome and those with the HiFi-based assembly (the assembly prior to “T2T”). Nonetheless, I think this section does not provide much novel information and can be removed.

RESPONSE: We agree with the reviewer suggestion and have removed this section.

Line 369-395: It's widely acknowledged that identification of CNVs based on short-read data presents a high false positive rate. To ensure the reliability of the identified CNVs, evaluation of the CNV accuracy is necessary. In addition, the list of these CNVs should be provided in a supplementary table.

RESPONSE: Thank you for the suggestion. We agree that detecting CNVs using NGS sequencing data has its limitation but it offers a rapid and effective way to estimate CNVs than experimental approach that is usually difficult and time-consuming. The approach has been routinely used in many genomic studies for human, animals and plants, using short-read sequencing data. In our study, we utilized the resequencing data of 311 public available accessions, and NGS data from 9 accessions generated by us to call CNVs. After mapping to CaT2T, we calculated the read coverage at the capsaicin biosynthesis genes (CBGs) and normalized them against the whole-genome sequencing coverage, based on which we calculated

the copy numbers of all CBGs in all 320 accessions, and recorded the CNVs in a table. Similar logic has been widely used in CNV detection in other genomic studies. We believe it suits our purpose for this study and the machine learning model has successfully predicted capsaicinoid levels using the detected CNVs of key CBGs, demonstrating its efficacy for genomic prediction in pepper breeding. Per your suggestion, we have added information on the detected CNVs of capsaicin biosynthesis genes in Supplementary Table 15.

It appears challenging to see from Fig. 6A that “For cultivars with intact CS gene, capsaicinoid levels appeared to be affected by CNVs of other CBGs” (Line 378). Statistical analyses are needed here.

RESPONSE: We draw this conclusion from combination of the CNV heatmap and line graph of capsaicin level (**Fig 5A**), where it is clear that samples with CS copy number ≥ 1 produce variable capsaicin levels. This of course requires a statistical support and it is why we trained a machine-learning model to capture the association of gene CNVs and capsaicin level. Please see our statement about this on Line 370-373: "To understand the relationship between CNV and capsaicinoid levels, we trained several machine-learning models including random forest, decision tree, K-nearest neighbor, Gaussian Naive Bayes, and SVM (support vector machine), taking CNVs and capsaicinoid levels as input."

To avoid confusion, we also revised this sentence to “For cultivars with intact CS gene, capsaicinoid levels might be affected by CNVs of other CBGs.” (Line 369-370)

Minor:

*Line 64 and 118: Genome sequences of wild peppers are available, e.g., *C. annuum* var. *glabriusculum* (<https://pnas.org/doi/full/10.1073/pnas.1400975111>).*

RESPONSE: We revised these two sentences as “..., whereas genome sequences for wild peppers are very scarce⁷.” at Line 67 and “..., the first gap-free genome for non-domesticated *Capsicum*” at Line 122.

Please add the accession name for *C. rhomboideum* that was used for genome assembly.

RESPONSE: We added the accession name of 'Andean' for *C. rhomboideum* in the revised manuscript (Line 101 and 478).

Fig. 1A panel b: the beginning of Chr08 of CaT2T has a very high GC content compared to other genomic regions. Have the authors checked what are the possible reasons that cause this?

RESPONSE: The high GC content at the beginning of Chr08 of CaT2T was due to the presence of 45S rDNA repeats.

Table S4: why LAI of CaT2T was much lower than CrT2T. The authors should check all other published pepper genomes to make sure that cultivated peppers indeed have a low LAI.

RESPONSE: Thanks for your comment. Indeed, Capsicum genomes tend to have low LAI which has been reported by a previous paper (<https://doi.org/10.1093/aobpla/plad015>) showing that Solanaceae plants (including *Capsicum*) LAI are typically low (mostly below 10) (https://academic.oup.com/view-large/figure/405476340/plad015_fig4.jpg). In fact, we downloaded several published high-quality *C. annuum* pepper genomes and calculated their LAI. The results showed the LAI of 8.87 for CC260 (contig N50=135.1 Mb) and 8.94 for CC090 (contig N50=187.1 Mb). Altogether, this suggests the *Capsicum* genomes generally have a characteristic low LAI for some reasons.

Line 280-282: As the authors stated, this could be “a result of gene loss”, then how this can highlight the importance of T2T assembly? Much deeper analysis is needed to reach such conclusion (e.g., confirm the genes were present in C. baccatum but not assembled).

RESPONSE: Thanks for your comments. We have updated this analysis using the latest version of *C. baccatum* and *C. pubescens* assemblies published recently (Liu *et al.* Nature communications 2023). The analysis did find four CS homologs in *C. baccatum* and *C. pubescens*, compared to seven

CS genes in *C. annuum*. The microsynteny analysis indicated the lower copies (four) of CS genes in *C. baccatum* and *C. pubescens* was due to either gene loss or additional rounds of tandem duplications in *C. annuum* during evolution.

Accordingly, we have revised the manuscript text as the following (Lines 292-295):

“Particularly, *Capsicum* species had the most copies of CS with seven in *C. annuum*, six in *C. chinense*, and four in *C. pubescens*, *C. baccatum* and *C. rhomboideum*. The *C. pubescens* and *C. baccatum* had fewer copies than *C. annuum*, due to either gene loss in the former two, or continuous tandem duplications in the latter”

Line 287-289: Please be careful to describe the results here. From Fig. 4D, it is clear that NOT all CBGs displayed high expression levels in pungent Capsicum species.

RESPONSE: Thanks for your suggestion. We have revised the ‘all’ to be ‘many’. (Line 300)

Line 295-296: The authors mentioned just above that SVs are present in CDS, making the statement here invalid.

RESPONSE: We appreciate your suggestion. We have revised this statement as “although non-pungent species contained homologs of functional CBGs, their coding and regulatory regions had diverged significantly from those of *C. annuum*.” (Line 309-310)

CNVs were identified for 75 of 124 CBGs (Line 375). It is confusing that how “the SVM model ranked 124 CBGs” (Line 387) as CNVs were used in the model while 49 CBGs do not harbor CNVs?

RESPONSE: Thank you very much for the kind suggestion. That is a mistake of writing from our side. In fact we only trained the model using CNVs of 75 CBGs, and ranked these 75 CBGs. We have revised the manuscript accordingly (Line 380-383):

"Furthermore, the SVM model ranked 75 CBGs by their contribution to prediction performance. The top-ranked genes included well known CBGs such as *CS*, *CCoAOMT*, *C3H*, *PAL*, *KasI*, *MYB31* and

putative ones as *SAMSyn*, *IPMS* and *HCT* etc. (Fig. 5D), majority of which were highly expressed in fruits (Supplementary Table 16)."

Line 411-412: Only four Capsicum genomes were used (Fig. 4A), instead of six.

RESPONSE: Thanks for your suggestion. We have revised the number of *Capsicum* genomes to five (four plus a new *C. pubescens* genome). (Line 405)

Reviewer #3 (Remarks to the Author):

This study assembled two T2T gapless Capsicum genomes for the pungent pepper C. annuum and its non-pungent wild relative C. rhomboideum. The authors examined distinct structural, epigenetic, and transcriptional features in their centromeres to delineate the evolution of fruit pungency in chili peppers. Specific chromatin regions associated with tissue-specific gene regulation were identified. The complete C. annuum genome improved variant detection and was used to create a machine learning model to predict pungency for genomic selection. However, I would like to suggest a substantial revision for this manuscript.

RESPONSE: Thank you very much for your valuable comments and suggestions helping us to improve the manuscript. We have carefully revised and improved the manuscript.

1. Genome validation. Although the authors reported T2T genome of two pepper species, more data to accept that those are really completed "Finished" T2T genomes. The authors should evaluate genome quality based on reliable evaluation process. The genome validation by the earth genome project can be helpful (<https://www.earthbiogenome.org/assembly-standards>). The authors need to confirm that those T2T genomes satisfy conditions for "Finished" suggested by the earth genome project.

RESPONSE: Thank you for the valuable comments on our manuscript. We have performed extensive validations of the two T2T genome assemblies using multiple approaches. First, we mapped HiFi, ONT and NGS reads against the T2T genomes and checked for read coverage for any

abnormal signals. For CaT2T, we found a mostly uniform coverage of mapped NGS. HiFi and ONT reads across whole genomes, while CrT2T generally had a uniform data coverage as well (Line 135-137) with a few exceptions due to the presence of high-copy satellite repeats (Supplementary Fig. 7). Reassuringly, such abnormal signals were associated with either HiFi or ONT mapping but not both at the same regions, suggesting the strength of each data type and highlighting the importance of combining both types for genome assembly to achieve accurate genome assembly. To show that gaps filled by combining different sequencing technologies, we added the coordinates of closed gaps in HiFi assembly compared to the final T2T assembly in Supplementary Table 3. Secondly, Hi-C data were mapped to both assemblies which revealed no structural assembly errors based on the chromatin interaction heatmap. Thirdly, to evaluate the assembly quality and completeness, we calculated the QV (quality value) and BUSCO score, which demonstrated high accuracy and completeness in two assemblies (Line 137-139). Based on our validations, we confirm that CaT2T is a *bona fide* finished genome containing all centromeres and telomeres (Line 117-120), whereas CrT2T is a nearly finished genome only missing a few telomeres (Line 121-124). In fact, we did not actually claim both genomes were finished, as modestly shown in the title of our manuscript "two telomere-to-telomere gap-free genomes".

Regarding the validation, the manuscript has been revised as the following (Line 131-143):

"We performed extensive validations of the two T2T genome assemblies. Firstly, we examined their Hi-C chromatin interaction maps showing no obvious misplacement of contigs within the CaT2T and CrT2T assemblies (Supplementary Fig. 7AB). Then we mapped all HiFi, ONT, and NGS reads separately against the assemblies, yielding a mapping rate of over 99.96% for all three data types (Supplementary Table 5). Mapped HiFi or ONT reads showed uniform coverage across all whole genome (Supplementary Fig. 7CD). The CaT2T and CrT2T had a QV (quality value) of 56.60 and 77.18, and a BUSCO score of 98.62% and 97.12%, respectively, demonstrating high accuracy and completeness of both assemblies (**Table 1**). Furthermore, aligning a recently published genome assembly of *C. annuum* cultivar '59' (hereafter Ca59)⁴ against CaT2T showed strong collinearity between the two assemblies (Supplementary Fig. 2D). The high-quality assembly of CaT2T was well supported by a high-coverage of HiFi and ONT read mapping spanning these gap regions (Supplementary Fig. 8; Supplementary Table 6)."

2. In addition, the previously reported genome assemblies were used to highlight the genome quality of the CaT2T (*C. annuum*) assembly, capsicinoid gene analyses and variation detections. However, the latest genome version was recently reported and thus should be used in the analysis. *C. annuum* and *C. baccatum* genome assemblies were recently released in 2023 (Nat. Comm). Genome comparison, analyses for pungency-related genes (genes in *C. baccatum*) and highlighting variation detection parts should be updated after including additionally those most recently published genomes.

RESPONSE: We agree with your suggestion. Thus, we have updated our comparative genomics analysis using the recently published genome assembly of *C. baccatum* and *C. pubescens*.

Accordingly, we have updated the **Figure 4** and modified the manuscript text as the following:

Line 292-293: " Particularly, Capsicum species had the most copies of CS with seven in *C. annuum*, six in *C. chinense*, and four in *C. pubescens*, *C. baccatum* and *C. rhomboideum*. "

Line 302-305: "Sequence alignment revealed that these syntenic CS copies (CS-1/CS-2) had conserved coding sequences (CDS) and upstream and downstream regulatory regions among pungent species, while non-pungent species had structural variations (SVs) within both CDS and flanking regions (**Fig. 4E**; Supplementary Fig. 20)."

3. Total number of genes are over 45000 in both pepper genomes. These are too many comparing to previous pepper genomes and other Solanaceae genomes. Sustainable validation of gene annotation should be performed in addition to BUSCO. BUSCO analyses can only prove partial accuracy. In particular, the authors should evaluate and analyze over 10000 genes specifically in two pepper genomes in this study.

RESPONSE: Per your suggestion, we have revised the genome annotation pipeline by only keeping gene models with support from all three sources of evidence (see below), which predicted 34,428 and 33,512 high-confidence protein-coding genes in CaT2T and CrT2T, respectively. These are high-confidence gene models supported by a combination of *ab initio* prediction, homology proteins and transcriptomic evidence (both NGS and Iso-seq). In addition, we have also performed

validation of the gene models in IGV-GSAman (<https://gitee.com/CJchen/IGV-sRNA>) to correct erroneous protein-coding genes. Therefore, our new genome annotation is of high-quality and reliable. The numbers of gene models are now comparable to previously published genomes and other Solanaceae genomes. BUSCO evaluation using solanales_odb10 database (n=5,950) indicated 97.04% and 93.23% orthologs were completely annotated in the updated annotation of CaT2T and CrT2T.

4. The authors used Hi-C data of CA59 to analyze TADs in CaT2T. However, the epigenomic structures such as TADs are different between i) accessions including between same accessions. I cannot agree that the authors have concluded their findings based on TAD analyses using CA59 for different accession as reference genome. To generalize, the authors should use Hi-C data for multiple accessions in addition to CA59 and should conclude their results.

RESPONSE: Thanks for your suggestion. We believe the reviewer misunderstood our TAD data analysis. In fact, we used our own generated Hi-C data of *C. annuum* to call TADs in this study, and mapped them against the CaT2T genome. Our conclusion about TAD is drawn using Hi-C data generated for *C. annuum* and analyzed using CaT2T genome in this study. We have revised the source of Hi-C data in the revised manuscript as the following:

Line 723-724: “Hi-C data was generated from leaf tissue as above mentioned, and processed using HiC-Pro (v3.1.0) and Juicertools (v1.22.01)”.

*5. Genome annotation predicted that the CaT2T and CrT2T centromeres contain 252 and 386 genes, respectively. The authors identified the divergence of centromere genes between two species and suggested their potential function based on the GO analysis. Although there are many centromere-encoded genes in *C. rhomboideum*, only RNA-seq data for *C. annuum* was used to describe transcriptional activity. Why were the centromere-encoded genes in CrT2T not identified?*

RESPONSE: We appreciate your suggestion. We have revised the manuscript by adding the expression of centromere-encoded genes in two *Capsicum* assemblies as Supplementary Table 11. And we have added the description of CrT2T centromeric genes as the following: “For example,

about 42 (70.0%) centromere-encoded genes were expressed in at least one tissue (TPM > 1) in *C. annuum*, including CaT2T07g00954 encoding telomere maintenance protein that protects the ends of telomeres from attack and CaT2T01g02835 encoding and flowering time control protein. The expression of CrT2T centromere-encoded genes (92.5%) was more active, containing three tandem copies of RCC1 (regulator of chromosome condensation) with an average TPM > 200, potentially playing key roles in the regulation of chromatin condensation in mitosis.” (Lines 235-241)

6. In *C. annuum*, centromere-located CRMs showed lower CHG methylation of gene-body, suggesting transcriptional activity of CRMs. Have you examined it for the other gypsy subfamilies such as *athila* and *tekay*?

RESPONSE: Thank you very much for the feedback. We have revised the manuscript to show the methylation level of centromere-located *Athila*, *CRM* and *Tekay* elements in Supplementary Fig. 17 as below. As we can see, the *CRMs* and *Tekays* showed lower or slightly lower of CHG methylation in gene-body, while *Athila* showed no significant difference between gene-body and flanking regions. (Line 252-255)

Supplementary Fig. 17. Methylation levels of centromere-located *Athila*, *Tekay* and *CRM* including the upstream and downstream 5 kb regions in *C. annuum*.

7. Sequence alignment of capsicinoid biosynthesis genes (CBGs), such as *ACL*, *BCAT*, *CCoAMT*, *FatA*, and *KasI*, showed that sequence variations were observed in *Capsicum* species. Moreover, it was identified that all CBGs were highly expressed in fruits of pungent *Capsicum* species, while hardly expressed in *C. rhomboideum* and *Physalis*. It is also well known that the biosynthesis of capsaicin was occurred in the fruit placenta tissue of pepper. Have these sequence variations and expressions for each tissue been experimentally validated? As I mentioned above, *C. baccatum*

genome was recently published and the authors need to check CS genes are really low copy in the high-quality C. baccatum genome.

RESPONSE: Thank you for your suggestion. The sequence variations were carefully analyzed using sequence alignment of the high-quality genome assemblies (including ours and public domains), whereas the gene expressions were analyzed using the tissue-specific RNA-seq data of chili peppers (this study) and other Solanaceae plants (downloaded from NCBI). Because we don't have the plant materials used to generate the public genome assembly and RNAseq data, it is difficult to experimentally validate the results involving these Solanaceae species. Nonetheless, the methods we used in this study are common approaches routinely taken in comparative genomics and transcriptomics, therefore our conclusions are reliable and provide important biological insights. Regarding the last comment, we have updated this analysis using the latest version of *C. baccatum* and *C. pubescens* assemblies published recently (Liu *et al.* Nature communications 2023). The analysis did find four *CS* homologs in *C. baccatum* and *C. pubescens*, compared to seven *CS* genes in *C. annuum*. The microsynteny analysis indicated the low copies (four) of *CS* genes in *C. baccatum* and *C. pubescens* was likely due to gene loss during evolution. We have revised this sentence as “Particularly, *Capsicum* species had the most copies of *CS* with seven in *C. annuum*, six in *C. chinense*, and four in *C. pubescens*, *C. baccatum* and *C. rhomboideum*. The *C. pubescens* and *C. baccatum* had fewer copies than *C. annuum*, due to either gene loss in the former two, or continuous tandem duplications in the latter.” (Lines 292-295)

8. *This study suggests that MYB31 and MYB48 regulate several key genes responsible for capsaicin biosynthesis in pepper and DNA methylation in specific regions of them was detected using bisulfite sequencing. It is known that the expression level of not only transcription factors but also structural genes, which were related to the capsaicin biosynthetic process, can be regulated by the hypomethylation or hypermethylation at the gene level. Have authors investigated DNA methylation and expression features of genes that might be co-expressed with MYB31 and MYB48 in pungent and non-pungent peppers? This analysis can help to understand the model for the Capsicum-unique and tissue-specific regulation of the capsaicin biosynthesis genes in the hot pepper.*

RESPONSE: Thank you for your suggestion. Because we only performed co-profiling of whole-genome bisulfite sequencing and RNA-seq on whole fruit of pungent *C. annuum* pepper, we can only compare the methylation and expression levels of different CBGs in the pungent pepper. The comparison showed that majority of CBGs and the MYB transcription factors had low DNA methylation levels at open chromatin regions (marked by ATAC-seq peaks) in fruit tissue, consistent with their high gene expression levels in fruit, suggesting the low DNA methylation is negatively correlated with the transcription level. Therefore, our analysis showed that the co-regulation of structural genes and transcriptional regulators in fruit is likely conferred by fruit-specific accessible chromatin and hypomethylation within such regions.

We revised the Supplementary Fig. 21 to include the results regarding the key CBGs and MYB transcription factors, and revised the manuscript text as the following (Lines 339-342):

"The placenta-specific open chromatin regions (OCRs) with low methylation levels were detected within 2 kb upstream of *CS-2*, *MYB31* and *MYB48*, while *CS-1* also showed OCRs in both pulp and seeds, suggesting that *CS-2* is likely the primary functional gene that contributes to placenta-specific synthesis of capsaicinoids (**Fig. 4F**; Supplementary Fig. 21). "

Minor

- *Table S1 genome size estimation of C. rhomboideum is calculated using 1.71Gb.*

RESPONSE: We have re-calculated the sequencing depth of *C. rhomboideum* using genome size of 1.71 Gb in Supplementary Table 1.

Reviewer #4 (Remarks to the Author):

This review focuses on the machine learning aspects of the genomic prediction in the paper.

RESPONSE: Thank you very much for your valuable comments and suggestions helping us to improve the manuscript. We have carefully revised and improved the manuscript.

To start, some of the comments here are only possible after reviewing the github code (notably

Gridsearch.py). Many of the details should be presented in the paper or supplementary material (e.g., hyperparameter ranges, etc.) The optimal hyperparameters from the grid search on the SVM must be presented in the paper (presumably they are RBF kernel with $C = 0.01$ and gamma the default in scikit-learn).

RESPONSE: Thank you for your insightful feedback. In response to your comments, we have revised the manuscript to include a new table that details the hyperparameters and their respective ranges for each machine learning model we evaluated. This addition can be found as Supplementary Table 17 in the revised manuscript. Moreover, we have explicitly reported the optimal hyperparameters for the SVM obtained from our grid search. Contrary to the presumption of an RBF kernel with $C = 0.01$, our findings actually indicate that the SVM with a linear kernel and a regularization parameter (C) set at 0.1 yielded the best results. This choice was made due to the interpretability benefits associated with a linear kernel, especially in terms of understanding the importance of input features.

Below is an overview of the hyperparameters and their ranges as included in the revised manuscript:

Supplementary Table 17: Hyperparameters and ranges for machine learning models

Model	Hyperparameters and Ranges
Support Vector Machine (Kernel: linear)	C : 0.0001 to 10
Random Forest	n_estimators: 10 to 150 max_depth: 1 to 30 max_features: None, sqrt, log2
K Nearest Neighbors	n_neighbors: 3 to 10 weights: uniform, distance
Gradient Boosting	n_estimators: 10 to 150 learning_rate: 0.01 to 0.2 max_depth: 1 to 30 max_features: None, sqrt, log2

We have revised the methods to reflect the changes we made as the following (Line 677-694):

"To train machine-learning models for capsaicin classification, we employed a balanced dataset created by dividing samples based on the median capsaicinoid content. The input features comprised CNVs of putative capsaicinoid biosynthetic genes, while the output was categorized into high and low capsaicinoid content based on the median value. Four primary algorithms—Random Forest, Support Vector Machines (SVM), Gradient Boosting, and K Nearest Neighbors (KNN)—were utilized to model the relationship between input features and classification outcomes. A 10-fold cross-validation approach was adopted for model evaluation, using implementations from the scikit-learn library. We applied a Grid Search approach to meticulously explore an extensive range of hyperparameters for each algorithm, aiming to identify the most effective model parameters. In the selection of kernel types for the SVM models, we focused on polynomial (poly) and linear kernels to enhance the interpretability of input features in subsequent analyses. Supplementary Table 17 provides a detailed overview of the models and their respective hyperparameter ranges. The SVM model, particularly with a linear kernel and a regularization parameter (C) set to 0.1, exhibited the highest accuracy, approximately 68%. This configuration achieved superior performance as evidenced by the ROC curve, attaining the highest AUC value of 0.84. This underscores its proficiency in classifying capsaicinoid content based on input CNVs of significant genes. Then the importance of input feature genes for capsaicinoid content were also evaluated with this SVM classification model."

The use of a binary classification framework rather than doing regression is not motivated, and is an unnecessary simplification. If there is a motivation for the binary classification in real-world terms? The authors should show the results of using an SVR to do prediction of the raw capsaicinoids content using the same hyperparameters as the classification problem, or preferably a new grid search for the regression problem. If this is not done, the authors should at a very minimum provide motivation for this choice of classification (presumably, because it is easier and gives better results).

RESPONSE: Thank you for your valuable input regarding our methodological choices. In the revised manuscript, we have provided a more detailed justification for our decision to employ a binary classification framework rather than a regression approach.

Our primary reason for choosing binary classification stems from the specific goals of our study and the characteristics of our dataset. Given the limited size of our dataset, a binary classification model proved to be more robust and less sensitive to noise, enhancing overall performance. This is particularly crucial in our context, where accurately distinguishing between high and low levels of capsaicinoids is more relevant than predicting their exact concentration.

Furthermore, the binary classification framework offers a clearer and more direct interpretation for our study's objectives. It simplifies the understanding of gene influence on capsaicinoid presence, which is essential for practical applications such as selective breeding in agriculture. This approach aligns closely with our aim to discern the genetic factors contributing to capsaicinoid content rather than quantifying it.

We appreciate your suggestion regarding the use of Support Vector Regression (SVR) for predicting the exact content of capsaicinoids. While we did not explore this in the current study, we acknowledge that it could be an interesting avenue for future research. However, for the scope of this paper, we focused on the binary classification approach due to its suitability for our dataset and research objectives.

The authors acknowledge the limitations of the training set size, but also employ 30-fold CV with their training. With a training set of 311 elements, this gives a test set of approximately ten in each fold. This is too small, creating the possibility of larger variation between folds. I would suggest a fold number in the range of 5-10 to verify the same results hold (Strangely, the github code reveals a cv parameter of 80. Surely this must not be the value that was used.) If the authors feel that 30-fold CV is required, which would be very unusual, there should be some citation to the literature as to the requirements, especially in the context of the small training set. In the absence of this, redoing the experiments with a CV with 5-10 folds would be necessary.

RESPONSE: Thank you for the suggestion. We understand your concern regarding the potential for increased variability due to the small test set size in each fold of our 30-fold cross-validation. We have re-evaluated our validation strategy and agree that a smaller number of folds is more appropriate given the size of our dataset. Consequently, we have revised our approach and conducted a 10-fold cross-validation, which is a widely accepted standard in machine learning practices and offers a balance between variance and bias in the validation process. This adjustment to a 10-fold cross-validation aligns better with the size of our training set (311 elements) and mitigates the risk of large variations between folds. It also ensures that each fold has a sufficiently large test set, enhancing the reliability and robustness of our model evaluation.

We have updated the code on GitHub to reflect the changes made in our study, switching from the previously mentioned 30-fold to a more suitable 10-fold cross-validation. This adjustment aligns with standard practices and enhances the reliability of our results. We have also re-run our analyses with this new configuration and updated our findings accordingly. The updated analysis does not change the main conclusion about the model and its performance. We appreciate your attention to detail in reviewing our code and are grateful for the opportunity to improve the accuracy and transparency of our work.

The manuscript has been revised regarding the updated analysis as the following (Line 373-375):
"With a 10-fold cross-validation, a SVM model (linear kernel, $C = 0.1$) achieved the highest average prediction accuracy of around 0.76 (**Fig. 5B**) and its ROC (receiver operating characteristic) curve showed the highest AUC (area under curve) value of 0.84 (**Fig. 5C**). "

The grid search for RBF without searching for optimal values of gamma is unusual. The authors should consider this hyperparameter in their grid search, or otherwise validate why this was not done.

RESPONSE: Thank you for pointing out the importance of considering the gamma hyperparameter in the grid search for an RBF kernel. We acknowledge that typically, optimizing both C and gamma

is essential for fine-tuning an RBF kernel-based SVM. However, our experimental results led us to choose a linear kernel for the SVM, primarily because of its interpretability benefits, particularly in understanding the importance of input features. With a linear kernel, the model's performance was not only comparable but also offered clearer insights, making it more suitable for our study's objectives. Consequently, we did not conduct an extensive grid search for the gamma parameter with the RBF kernel, as our focus shifted to leveraging the linear kernel's advantages. We believe this decision aligns well with our study's goals and provides a more straightforward interpretation of the results.

Code for the Decision tree, Random forest or KNN is not included in github and insufficiently described in the paper. The range of hyperparameters that were considered for each of these models should at a minimum be described somewhere.

RESPONSE: Thank you for your valuable feedback. We understand the importance of transparency and thorough documentation in our research. In response to your comments, we have updated our manuscript to include a comprehensive new table (Supplementary Table 17) that details the hyperparameters and their respective ranges for each machine learning model, including the Decision Tree, Random Forest, and KNN models. This additional information is now clearly presented in the revised manuscript.

Furthermore, to ensure completeness and aid in reproducibility, we have also updated our GitHub repository to include the code for the Decision Tree, Random Forest, and KNN models. This will provide a more in-depth understanding of our implementation and enable others to replicate or build upon our work more easily. Please find these amendments and the associated code in the updated GitHub repository (https://github.com/Weikai-47/Pepper_T2T/tree/main/SVM-based%20classifier), with detailed comments added to enhance clarity.

We greatly appreciate your constructive feedback and trust that our revisions address your concerns effectively. Thank you for your invaluable contribution to our work.

Reviewers' Comments:

Reviewer #1:

Remarks to the Author:

Thank you for adding the requested information. The manuscript is much improved and represents a valuable contribution to the field of plant genomics.

Reviewer #2:

Remarks to the Author:

The authors have addressed most of my concerns. Here are several remaining ones (most are minor while the first one is outstanding):

CNV detection: The authors used a very simple method to detect CNVs, according to the text in Methods (Line 675-676: "For CNV detection, read depths were calculated in 1 Mb windows or per gene using Samtools coverage and normalized by the overall median read depth"). This would bring a lot of false calls that could be caused by library construction and sequencing artifacts, read mapping errors etc. In addition, the CNV results shown in Table S15 do not make sense. How can the copy number be something like "1.312987316", "0.864030844", "0.511303024"...? It should be zero, one, two copies..., and may include homozygous or heterozygous... The authors mentioned in their response letter that "Similar logic has been widely used in CNV detection in other genomic studies". It would be great if the authors can provide several example references.

The authors named the accession "C. rhomboideum cv. Andean" (Line 101). First, "cv." should not be used here as this is a wild accession. Second, does this accession has a PI number? Where did the authors obtain this accession? "Andean" seems to be a weird accession name.

The authors provided a table listing the gaps closed in the final T2T genomes (Table S3). What is the meaning of lengths with negative values in this table?

Line 143-146 and Fig. S9: There could be chimeric reads in ONT sequencing, which would misjoin nuclear and chloroplast/mitochondrion sequences in the assembly. For the example shown in Fig. 9 (and others), are there any HiFi reads that support the two junction sites between the nuclear and chloroplast/mitochondrion sequences in CaT2T, and HiFi and ONT reads that support the assembly of Ca59 in this region (spanning the break point)?

Line 237-238: Gene names should be italic. Change "encoding telomere maintenance protein" to "encoding a telomere maintenance protein", and "encoding and flowering time control protein" to "encoding a flowering time control protein".

Line 295: What is the exact meaning of "continuous tandem duplications"?

Line 443: The CS2 gene was also expressed in pulp; therefore, "placenta-specific transcription" here is not accurate.

Line 774: In the database, for CrT2T, only the genome assembly is made available. Please also add the access to the gff3, cds and protein files of CrT2T.

Reviewer #3:

Remarks to the Author:

I am happy to verify the revised manuscript and the authors have addressed my concerns

Reviewer #4:

Remarks to the Author:

Thank you for considering the comments and updating the paper as appropriate. I am satisfied with all of the rationale and changes made, especially the move to 10-fold CV.

My only suggestion is that the comments on classification vs regression are valid in their own right and might be considered for inclusion in the paper itself. In particular, the issue of small data set size and that, as you note in the reply, classification "simplifies the understanding of gene influence on capsaicinoid presence, which is essential for practical applications such as selective breeding in agriculture" is relevant to readers, especially those coming from a background in genomic prediction, where regression is more common (e.g., using BLUP or similar techniques).

Reviewer #1 (Remarks to the Author):

Thank you for adding the requested information. The manuscript is much improved and represents a valuable contribution to the field of plant genomics.

RESPONSE: Thank you very much for your consideration.

Reviewer #2 (Remarks to the Author):

The authors have addressed most of my concerns. Here are several remaining ones (most are minor while the first one is outstanding):

CNV detection: The authors used a very simple method to detect CNVs, according to the text in Methods (Line 675-676: “For CNV detection, read depths were calculated in 1 Mb windows or per gene using Samtools coverage and normalized by the overall median read depth”). This would bring a lot of false calls that could be caused by library construction and sequencing artifacts, read mapping errors etc. In addition, the CNV results shown in Table S15 do not make sense. How can the copy number be something like “1.312987316”, “0.864030844”, “0.511303024”...? It should be zero, one, two copies..., and may include homozygous or heterozygous... The authors mentioned in their response letter that “Similar logic has been widely used in CNV detection in other genomic studies”. It would be great if the authors can provide several example references.

RESPONSE: Thank you for your valuable suggestion for improving our manuscript. We have re-detected the CNVs in pepper accessions using whole-genome sequencing data by applying a published CNV calling pipeline called AMYCNE (Automatic Modeling functionality for Copy Number Estimation) (Eisfeldt et al. *PLoS One*, 2018), which has shown its superior performance to mainstream CNV detection tools such as CNVnator according to the developer (Eisfeldt et al. *PLoS One*, 2018). The CNVs detected for human amylase genes by AMYCNE was validated to be concordant to the ddPCR results, demonstrating its high accuracy in CNV detection (Eisfeldt et al. *PLoS One*, 2018). Thus, we applied this tool in the pepper datasets and the detected copy numbers for CBG genes were updated in Table S15 and Figure 5A. We then used the updated CNV results to re-train and test the machine-learning models, observing that the Random Forest model had the best performance of all trained models, with a prediction accuracy of 0.72, and a AUC of 0.85 (Figure 5BC) using a testing dataset. Overall, using the new CNV results, we obtained the machine-learning model with slightly improved performance compared to previous results.

We have revised the manuscript text accordingly as the following:

Line 364-366:

“Using CaT2T as reference genome, we detected genomewide CNVs from resequencing data of 311 *C. annuum* accessions with capsaicinoid quantification using AMYCNE (Supplementary Table 14)”

Lines 373-377:

“After performing recursive feature selection and grid search, a Random Forest model achieved the highest average prediction accuracy of approximately 0.72 using 10-fold cross-validation on a dataset composed of CNV detection from newly generated resequencing data with approximately 50×coverage (Table S14). The ROC curve of the Random Forest model also displayed the highest AUC value of 0.85 on this same dataset (Fig. 5C), demonstrating the potential of the model to accurately classify capsaicin levels from CNVs for genomic prediction at an early seedling stage.”

Lines 660-663:

“AMYCNE was employed to investigate copy number variations (CNVs) of putative capsaicin biosynthesis genes from whole genome re-sequencing reads, including 311 publicly available accessions of *C. annuum* (BioProject accession: PRJCA004361) (for model training) and nine additional accessions sequenced in this study (for model testing).”

Lines 689-692:

“The random forest model with following parameters “max_depth=20, max_features='sqrt' and n_estimators=150” exhibited the highest accuracy, approximately 72%. This configuration achieved superior performance as evidenced by the ROC curve, attaining the highest AUC value of 0.85. ”

The AMYCNE citation:

Eisfeldt J, Nilsson D, Andersson-Assarsson JC, Lindstrand A (2018) AMYCNE: Confident copy number assessment using whole genome sequencing data. *PLoS ONE* 13(3): e0189710.

<https://doi.org/10.1371/journal.pone.0189710>

The authors named the accession “*C. rhomboideum* cv. *Andean*” (Line 101). First, “cv.” should not be used here as this is a wild accession. Second, does this accession has a PI number? Where did the authors obtain this accession? “*Andean*” seems to be a weird accession name.

RESPONSE: Thank you for your suggestion. We have examined our record and found that the *C. rhomboideum* accession did have a PI number. Therefore, we have revised the ‘cv. *Andean*’ to ‘wild accession PI 645680’. (Lines 101-102, 477-478). This accession was originally obtained from Fatalii Seeds Supplier. We apologize for the omission of the accession information previously.

The authors provided a table listing the gaps closed in the final T2T genomes (Table S3). What is the meaning of lengths with negative values in this table?

RESPONSE: Thank you for your comment. We classified the gaps of HiFi assembly in two types as follows (detailed diagram in Table S6). The lengths were expressed as ‘a – b’ and the negative values meant the ‘overlap’-type gaps.

Line 143-146 and Fig. S9: There could be chimeric reads in ONT sequencing, which would misjoin nuclear and chloroplast/mitochondrion sequences in the assembly. For the example shown in Fig. S9 (and others), are there any HiFi reads that support the two junction sites between the nuclear and chloroplast/mitochondrion sequences in *CaT2T*, and HiFi and ONT reads that support the assembly of *Ca59* in this region (spanning the break point)?

RESPONSE: Thank you for your suggestion. We agree with you that chimeric ONT read is possible. Therefore, we have examined the HiFi read mapping in IGV, and checked the HiFi reads that covered the two junction sites between the nuclear and chloroplast sequences in *CaT2T* through IGV, which supports the assembly of multiple chloroplast/mitochondrion insertions. We have included the HiFi coverage in Fig. S9 as an example. The analyses on the *Ca59* assembly were only conducted on the public released genome assembly, thus we could not evaluate *Ca59* assembly quality as we did for *CaT2T*.

Line 237-238: Gene names should be italic. Change “encoding telomere maintenance protein” to “encoding a telomere maintenance protein”, and “encoding and flowering time control protein” to “encoding a flowering time control protein”.

RESPONSE: Thank you for your suggestion. We have revised the gene names to be italic. (Lines 237, 238, 383)

Line 295: What is the exact meaning of “continuous tandem duplications”?

RESPONSE: Thank you for your comment. The higher copy number of CS in *C. annuum* than *C. pubescens* and *C. baccatum* was probably due to continuous expansion of CS copies through tandem duplications.

Line 443: The CS2 gene was also expressed in pulp; therefore, “placenta-specific transcription” here is not accurate.

RESPONSE: Thank you for your suggestion. We have revised the “placenta-specific transcription” to “transcription”. (Line 442)

Line 774: In the database, for CrT2T, only the genome assembly is made available. Please also add the access to the *gff3*, *cds* and protein files of CrT2T.

RESPONSE: Thank you for your suggestion. We have now included the annotation files of CrT2T

in the pepper-site database (<http://www.pepperbase.site/node/3>).

Reviewer #3 (Remarks to the Author):

I am happy to verify the revised manuscript and the authors have addressed my concerns.

RESPONSE: Thank you very much for your valuable suggestions.

Reviewer #4 (Remarks to the Author):

Thank you for considering the comments and updating the paper as appropriate. I am satisfied with all of the rationale and changes made, especially the move to 10-fold CV. My only suggestion is that the comments on classification vs regression are valid in their own right and might be considered for inclusion in the paper itself. In particular, the issue of small data set size and that, as you note in the reply, classification "simplifies the understanding of gene influence on capsaicinoid presence, which is essential for practical applications such as selective breeding in agriculture" is relevant to readers, especially those coming from a background in genomic prediction, where regression is more common (e.g., using BLUP or similar techniques).

RESPONSE: Thank you very much for your valuable suggestions. We have now included the comments in our manuscript as the following:

“The primary reason for choosing binary classification of capsaicinoid content stems from the specific goals of our study and the characteristics of our dataset. Given the limited size of our dataset, a binary classification model proved to be more robust and less sensitive to noise.

Furthermore, the binary classification framework offers a clearer and more direct interpretation for our objectives. It simplifies the understanding of gene influence on capsaicinoid presence, which is essential for practical applications such as selective breeding in agriculture. This approach aligns closely with our aim to discern the genetic factors contributing to capsaicinoid content rather than quantifying it.”. (Lines 667-674)

Reviewers' Comments:

Reviewer #2:

Remarks to the Author:

The authors have addressed all my minor concerns; however, I am still not convinced by their results from the CNV detection. By checking Table S15, I found that most genes have five, six, or seven copies in most of these accessions (an average of 5.97 copies per gene across the table), which seems to not make sense and is highly inconsistent with the normalized read depth in Table S15 of their previous version (an average read depth of 0.95 across the table). I then checked the paper describing the method (AMYCNE) the authors used for CNV detection and found out that in that paper, the performance of AMYCE was only evaluated using one locus in humans, AMY1. This goes back to my comment during the first round of my review: "It's widely acknowledged that identification of CNVs based on short-read data presents a high false positive rate. To ensure the reliability of the identified CNVs, evaluation of the CNV accuracy is necessary." Therefore, I suggest the authors find a more robust and thoroughly evaluated tool for their CNV detection and may present some manual checking results based on the read alignments to validate their detected CNVs.

Reviewer #2 (Remarks to the Author):

The authors have addressed all my minor concerns; however, I am still not convinced by their results from the CNV detection. By checking Table S15, I found that most genes have five, six, or seven copies in most of these accessions (an average of 5.97 copies per gene across the table), which seems to not make sense and is highly inconsistent with the normalized read depth in Table S15 of their previous version (an average read depth of 0.95 across the table). I then checked the paper describing the method (AMYCNE) the authors used for CNV detection and found out that in that paper, the performance of AMYCE was only evaluated using one locus in humans, AMY1. This goes back to my comment during the first round of my review: "It's widely acknowledged that identification of CNVs based on short-read data presents a high false positive rate. To ensure the reliability of the identified CNVs, evaluation of the CNV accuracy is necessary." Therefore, I suggest the authors find a more robust and thoroughly evaluated tool for their CNV detection and may present some manual checking results based on the read alignments to validate their detected CNVs.

RESPONSE:

Thank you very much for your comment and suggestion. Following your suggestions, we revisited the evaluation of CNV detection using diverse tools. We undertook a comparative analysis of CNVs and their consistency across samples with varying sequencing depths. Additionally, we examined the lengths of CNV segments identified by various CNV detection tools in addition to previously used tool AMYCNE. We employed DELLY (Rausch et al., Bioinformatics, 2012), LUMPY (Layer et al., Genome Biology, 2014), CNVpytor (Suvakov et al., Gigascience, 2021), and Vaquita (Kim and Reinert, WABI, 2017) to detect CNVs in 320 samples, encompassing 311 low-depth (average depth of 8.6x) sequencing samples (Cao et al. 2022) and 9 high-depth (average depth of 60x) sequencing samples generated by this study.

The results (**Fig. R1**) revealed a significantly smaller number of CNVs compared to those obtained using AMYCNE, highlighting the instability of detection among different tools when utilizing the same samples. Within the low-depth group, the median copy number (marked by the red dashed line) detected by DELLY, LUMPY, CNVpytor, and Vaquita was 2, indicating their difficulty in detecting copy number variations. Conversely, in the high-depth group, the median copy number detected by LUMPY and Vaquita was 3, suggesting a higher likelihood of detecting CNVs. Furthermore, a reduction in outliers was observed in the results obtained from AMYCNE and CNVpytor. Therefore, a deeper sequencing depth tends to enhance chances of detecting more reliable CNVs.

Figure R1. Distribution of detected CNV numbers using five software in samples of low and high sequencing depths.

Subsequently, we compared the consistency of copy number detection across different tools on two different groups of samples: one with 9 high-depth samples and another with 9 low-depth samples. Among all five tools, a total of 3,522 CNVs were identified, and none of them exhibited complete consistency across all tools in the low-depth group (**Fig. R2a**). In contrast, 4,300 CNVs were detected in the high-depth group, with 8 of them exhibiting complete consistency across all tools (**Fig. R2b**). This suggests that as sequencing depth increases, both the detection rate of CNVs and their consistency among detectors improve. Consequently, utilizing high-depth sequencing samples enables us to identify more robust and reliable CNVs, whereas low-depth samples may not yield such reliable results.

Figure R2. Concordance of CNVs detected with low (a) and high (b) sequencing depth samples.

Furthermore, we investigated the distribution of CNV variant lengths detected by each tool and compared them with the lengths of capsaicin biosynthesis genes (CBGs). Our observations revealed that the majority of CNV lengths detected by DELLY, LUMPY, and Vaquita were shorter than 10K, whereas CNVpytor primarily detected longer variants exceeding 10K (Fig. R3a). Conversely, the lengths of most CBGs were primarily below 10K (Fig. R4). In a subsequent analysis, we examined the total amount of genes exhibiting copy number variations across the 320 samples. Notably, DELLY, LUMPY, and Vaquita detected a higher frequency of genes with copy number variations, whereas CNVpytor detected much fewer instances (Fig. R3b). This may be related with the distribution of pepper gene lengths and the CNV lengths detected by different tools.

Figure R3. Distribution of CNV segment length (a) and the total count of detected CNVs (b) among samples detected by different tools.

Figure R4. Distribution of length of capsaicin biosynthesis genes.

Based on these analyses, it is evident that sequencing depth has a profound impact on our CNV detection, and low-depth short-read data particularly poses challenges in obtaining robust and reliable results, as demonstrated by results from tools employed above which all failed to generate credible CNVs of the CBGs. In fact, majority of our short-read data was obtained from public databases with low sequencing depth (average 8.6x) released by Cao *et. al* (2022).

Therefore, considering your concern about the CNV detection method (rightly so), and the general limitation of CNV detection using low-coverage short-read data, as well as its dispensableness to this study (see below for justification), we think it is best for us to remove this section about CNV and the subsequent machine-learning model to ensure the rigor of our conclusions, and make the manuscript suitable for publication. Since this section represents only a small side topic dispensable to the manuscript, we believe that removing it from the manuscript will not affect the overall conclusion and the entirety of our study.

The major contributions of our breakthrough study include generating the first two T2T gapless genome assemblies of *Capsicum* species, conducting in-depth analyses on the mysterious genetic and epigenetic landscape of centromeres, and revealing the evolutionary insights into the capsaicinoid biosynthesis pathways. The timely genomic resources and biological insights from these analyses already represent a major advancement in the field. As for the CNV analysis and model training, we meant to demonstrate it as one example of the potential applications of such genomic resources. However, given your concern about the analysis and other consideration mentioned above, we believe it is better to exclude it from the manuscript and investigate the issue in future studies. This analysis is dispensable, and removing it does not affect the main

conclusions of our manuscript such as the genome assembly, centromere findings, epigenetic regulation of capsaicinoid biosynthetic pathway etc.

Therefore, we decided to remove the section relevant to the CNV calling and machine-learning from the manuscript. The revised manuscript still includes six main display items (5 figures + 1 table), which is common for a typical research article.

We very much appreciate your understanding and consideration. Thank you again for your dedication and great suggestions throughout the review process to improve our manuscript.

References:

Cao, Y. et al. Pepper variome reveals the history and key loci associated with fruit domestication and diversification. *Mol. Plant* **15**, 1744-1758 (2022)

Rausch, T. et al. DELLY: structural variant discovery by integrated paired-end and split-read analysis. *Bioinformatics* **28**, i333-i339 (2012)

Layer, R. M. et al. LUMPY: a probabilistic framework for structural variant discovery. *Genome biology* **15**, 1-19 (2014)

Suvakov, M. et al. CNVpytor: a tool for copy number variation detection and analysis from read depth and allele imbalance in whole-genome sequencing. *Gigascience* **10**, giab074 (2021)

Kim, J., & Reinert, K. Vaquita: Fast and Accurate Identification of Structural Variation Using Combined Evidence. In *17th International Workshop on Algorithms in Bioinformatics (WABI 2017)*. Schloss Dagstuhl-Leibniz-Zentrum fuer Informatik (2017)

Reviewers' Comments:

Reviewer #2:

Remarks to the Author:

The authors addressed my concern related to CNV detection by removing the related content from the manuscript. This is fine with me.

Reviewer #2 (Remarks to the Author):

The authors addressed my concern related to CNV detection by removing the related content from the manuscript. This is fine with me.

RESPONSE:

Thank you very much for your comment and suggestion for improving our manuscript.